# Counterfactual Maximum Likelihood Estimation for Training Deep Networks

**Xinyi Wang, Wenhu Chen, Michael Saxon, William Yang Wang**
Department of Computer Science
University of California, Santa Barbara
`xinyi_wang@ucsb.edu, wenhuchen@ucsb.edu, saxon@ucsb.edu, william@cs.ucsb.edu`

## Abstract

Although deep learning models have driven state-of-the-art performance on a wide array of tasks, they are prone to spurious correlations that should not be learned as predictive clues. To mitigate this problem, we propose a causality-based training framework to reduce the spurious correlations caused by observed confounders. We give theoretical analysis on the underlying general Structural Causal Model (SCM) and propose to perform Maximum Likelihood Estimation (MLE) on the interventional distribution instead of the observational distribution, namely Counterfactual Maximum Likelihood Estimation (CMLE). As the interventional distribution, in general, is hidden from the observational data, we then derive two different upper bounds of the expected negative log-likelihood and propose two general algorithms, Implicit CMLE and Explicit CMLE, for causal predictions of deep learning models using observational data. We conduct experiments on both simulated data and two real-world tasks: Natural Language Inference (NLI) and Image Captioning. The results show that CMLE methods outperform the regular MLE method in terms of out-of-domain generalization performance and reducing spurious correlations, while maintaining comparable performance on the regular evaluations.[1]

## 1 Introduction

Deep neural networks have been tremendously successful across a variety of tasks and domains in recent years. However, studies have shown that deep learning models trained with traditional supervised learning framework tend to learn *spurious correlations* as predictive clues [1–4]. For example, in computer vision, deep learning models can rely on surface-level textures [1, 5] or background environment [2, 6] instead of the object of interest in images. In natural language processing (NLP), question-answering models are insensitive to the choice of question [7] and natural language inference models are surprisingly accurate at predicting the logical relationship between a pair of sentences from just one of them [3]. In image captioning, a phenomenon called *object hallucination* is observed where models tend to include nonexistent objects in a caption based on their common association with other objects that are actually present in the image [4].

Another related problem in the current associative learning framework, like maximum likelihood estimation (MLE), is that neural networks can achieve almost perfect performance on one dataset but dramatically fail to generalize on another because of a naturally occurring or adversarially enforced distribution shift [8–11]. One explanation is that the model learns 'fake' features induced by spurious correlations instead of invariant features that would hold in different domains for the same task [12]; this kind of invariance can be explained by the underlying causal mechanism of the task [13].

---

[1]Our code is released at `https://github.com/WANGXinyiLinda/CMLE`.

35th Conference on Neural Information Processing Systems (NeurIPS 2021).

It is usually impractical to identify and disentangle all of the potential sources of spurious correlations. Various training algorithms have been proposed to reduce the influence of spurious correlations by learning the underlying causal mechanism. One approach for discovering the underlying causal mechanism is by invariant prediction based on the invariance of causal mechanisms across different environments [13, 12, 14]. However, constructing such diverse environments is usually unrealistic. Another approach is counterfactual data augmentation [15–19], which directly modifies the part of the input that causes the target variable, but usually involves expensive human efforts in generating the counterfactual examples. We propose **Counterfactual Maximum Likelihood Estimation** (CMLE), which tries to perform MLE on an interventional distribution instead of the observational distribution. Here, **obervational distribution** means the distribution that the train data is sampled from. **Interventional distribution** means the distribution with one or some of the variables intentionally intervened (set to some fixed value). More specifically, we aim to remove observable confounders and reduce spurious correlations by directly intervening on the label variable, without requiring human annotators or the curation of diverse environments.

In this paper, we focus on the setting of predicting the outcome $Y$ of a certain action $T$ on $X$, with the underlying causal model as $X \rightarrow Y \leftarrow T$ and an observed train dataset in the form of $(X, Y, T)$. Such a setting enables us to assume that there exist some observed confounders in $X$ that influence both $Y$ and $T$. This means there is potentially a false causal relation from $X$ to $T$, indicated by a red arrow in Figure 1a. In our framework, instead of trying to identify all the confounders, we try to rule out the effect of the confounders by considering the interventional distribution that directly deleting the false causal edge from $X$ to $T$ with a resulting causal graph as shown in Figure 1d. This causal graph can be viewed as a Randomized Controlled Trial (RCT), which is the gold standard approach for estimating the treatment effect [20].

Our method is inspired by individual treatment effect (ITE) prediction [21, 22], which tries to predict the expected effect $\mathbb{E}[Y_1 - Y_0 | X = x]$ (e.g. differnce in blood pressure) of a treatment $T \in \{0, 1\}$ (e.g. drug/surgery) on a individual unit $X = x$ (e.g. a patient). In this case, the observed confounder in $X$ can be explained by selection bias [22], which means the treatment $T$ applied to an individual $X$ is dependent on $X$. For example, young patients are more likely to be treated by surgery, while elder patients are more likely to be treated by drugs.

There are two broad kinds of tasks to which our proposed CMLE framework can apply: relationship prediction tasks that predicts the relation $T$ given two inputs $X$ and $Y$ (e.g. natural language inference [23, 24], paraphrase identification [25, 26], natural language for visual reasoning [27], etc.), and conditional generation tasks that generate $Y$ given precondition $X$ and constraint $T$ (e.g. style transfer [28, 29], controllable image/text generation [30–32], controllable image captioning [33, 34] etc.). For conditional generation tasks, the causal relation of $X, Y, T$ fits in our setting. For relation prediction tasks, we first assume there is an underlying conditional generation process of $Y$ for a given pair of $X$ and $T$. This is a natural assumption as this is usually how the datasets for this type of task is generated. Then we augment the original dataset using the counterfactual examples generated by our method. In this paper, we try to reduce the spurious correlations contained in both kinds of tasks.

Since we generally cannot observe the interventional distribution, we derive two different upper bounds of the expected interventional negative log-likelihood using only the observational distribution: one is Implicit CMLE, as it does not involve any explicit generation of counterfactual examples; while the other is Explicit CMLE, as it explicitly generates counterfactual examples during the training process. We test our framework with deep neural networks on a simulated dataset and two real-world tasks with well-known spurious correlations: natural language inference [23] and image captioning [35]. Compared to regular MLE, we improve the out-of-domain accuracy of NLI on a hard adversarial dataset by 2.9% relatively and beat baselines on human preference evaluations by more than 10%, while maintaining comparable performance on automated evaluations. Our results show that our learning framework can better capture the underlying causal mechanism and reduce spurious correlations without degrading performance.

## 2   Related Work

A growing body of work has investigated algorithmic improvements for machine learning model training by leveraging underlying causal mechanisms. One line of work tries to utilize the invariance of causal mechanisms across different environments. Invariant Causal Prediction (ICP) [36] aims to

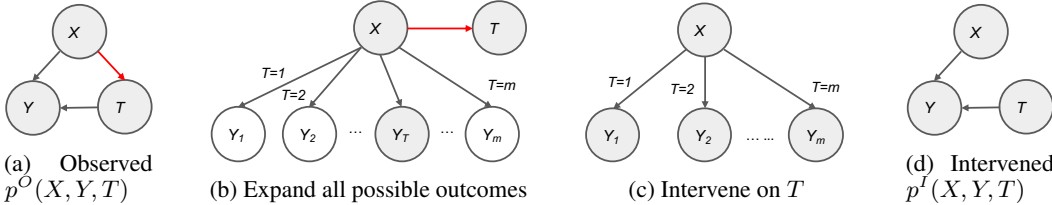

(a)   Observed   $p^O(X, Y, T)$     (b) Expand all possible outcomes     (c) Intervene on $T$     (d)   Intervened   $p^I(X, Y, T)$

Figure 1: The Structural Causal Model (SCM) assumed by Counterfactual MLE, where (a) and (b) have the same distribution, (c) is the resulting causal graph of intervening on $T$. (d) is the intervened distribution by only removing the $X \to T$ edge from (a). Shaded nodes means being observed and red arrows denote the causal relation causing the spurious correlation.

learn a linear invariant causal predictor to identify the causal parents of given target variables with data obtained from different environments. Further refinements on this line of work include nonlinear and nonparametric ICP assessment methods [37] and Invariant Risk Minimization (IRM) where out-of-distribution generalization is enabled by learning a nonlinear invariant causal predictor shared across different environments [12]. However, these methods generally require a set of environments that are both sufficiently diverse to eliminate confounders while maintaining the underlying causal structure. Though this can be achieved by intervening in the non-causal variables in a synthetic or interactive setting, it is very difficult to obtain or even define such environments with complex high-dimensional observational data like text or images. Our proposed CMLE framework only requires a single observational dataset, making it much more practical.

Another line of work approaches this problem with data augmentation. Methods including programmatically editing text to address gender bias [16–18], employing crowd workers to augment training data to capture potential unmeasured variables [38], and manually editing minimal text spans to flip the predicted label [15] have all been proposed to produce counterfactually augmented data (CAD) for improving out-of-domain generalization. Conceptually, in contrast to the first approach that involves interventions on a potentially large number of non-causal variables, CAD directly intervenes on a target variable [39], which is more practical for a specific task. Refinements on this work attempt to better utilize the manually edited or synthetically generated counterfactual data by pairing the factual and counterfactual examples together [40, 41], and leverage local causal structures to generate counterfactual experience in training reinforcement learning algorithms [19]. While human annotators can generate high-quality counterfactual examples that are effective in reducing spurious correlations, they are expensive to obtain and do not easily scale. Meanwhile, our proposed CMLE framework is fully automated, requiring no human annotations.

Our CMLE framework is inspired by a technique for predicting the Individual Treatment Effect (ITE) from observational data [22]. This method only observes one possible outcome of an individual $X = x$ with one binary treatment $T = 0$ or $T = 1$, and it assumes there are no hidden confounders but potentially observed ones. Assuming a similar underlying causal model as ours ($m = 2$) gives an error bound as the sum of the standard generalization error and the distance between the control and treated distributions in a representation space. Improvements on this work include disentangling the observable confounders in ITE using representation learning [42], and estimating the causal effects of linguistic properties [43]. While in this work we do not aim to separate the confounders from the causal factors in $X$, we try to reduce the spurious correlations caused by the possible confounders contained in $X$ using our proposed CMLE training framework.

## 3   Method

In this section, we first introduce our problem setting and the goal of Counterfactual Maximum Likelihood Estimation (CMLE). As we cannot observe the counterfactual outcomes in observational data, we propose two different ways of estimating the counterfactual likelihood by deriving two different upper bounds: Implicit CMLE, which is faster in training, and Explicit CMLE, which is more flexible. We also briefly introduce the deep learning architecture we use in our experiments.

### 3.1   Problem Setting

We consider the scenario with one input variable $X \in \mathcal{X}$ that contain all the confounders, one discrete treatment variable $T \in \{1, 2, ..., m\}$ where $m \geq 2$, and one outcome variable $Y \in \mathcal{Y}$,

as shown in Figure 1a. Note that $\mathcal{X}$ and $\mathcal{Y}$ can be very complicated high dimensional space like text or image and all three variables are observed in the dataset. Suppose the data we observed is $\mathcal{D} = \{(x_1, t_1, y_1), (x_2, t_2, y_2), ..., (x_n, t_n, y_n)\}$ and only $x_i$'s are sampled i.i.d. In this paper, we adopt the causal notations from [44]. Below we give a formal definition of Structural Causal Model (SCM) [44] that we use to model our framework:

**Definition 1.** *A Structural Causal Model (SCM) is a pair $\langle M, p(u) \rangle$, where $M$ is a triple $\langle U, V, F \rangle$ and $p(u)$ is a probability function defined over the domain of $U$. $U = \{U_1, U_2, ..., U_n\}$ is a set of background variables (exogenous) that are determined by factors outside the model. $V = \{V_1, V_2, ..., V_n\}$ is a set of variables (endogenous) and $F = \{f_1, f_2, ..., f_n\}$ is a set of functions s.t. $V_i = f_i(Pa(V_i), U_i), \ i = 1, 2, ..., n$, where $Pa(V_i) \in V \backslash V_i$ are the parents of $V_i$. We say that "$V_j$ is a direct cause of $V_i$" if $V_j \in Pa(V_i)$ and illustrate this relation as $V_j \rightarrow V_i$ in a causal graph.*

In Figure 1, we omit the exogenous and only show the causal relations between endogenous variables. The shaded nodes indicate that the corresponding variables are observed. Intervention on a specific variable is formalized as fixing the value of this variable to a specific value. This is known as the *do*-operation, changing the causal graph by removing all edges incident to the intervened variable. We denote the outcome of $Y$ corresponding to $do(T = t)$ as $Y_t$. We expand all the possible outcomes of $Y$ in Figure 1b while we actually can only observe one outcome for each data point in the observational dataset. We also make the following standard assumption [22, 45–49]:

**Assumption 1.** *(Strong Ignorability Condition) $(Y_1, Y_2, ..., Y_m) \perp\!\!\!\perp T | X$ and $0 < p(T|X) < 1$.*

Under this condition, the counterfactual outcomes are identifiable since $p(Y_t = y | X = x) = p(Y = y | X = x, T = t)$. i.e. The counterfactual likelihood can be written as observable conditional probabilities estimated from data.

Our goal is to eliminate the false causal relation from $X$ to $T$ represented by the red arrow in Figure 1a. To achieve this goal, we define our objective over the interventional distribution $p^I(X, Y, T)$ as shown in Figure 1d, instead of over the observed data distribution $p^O(X, Y, T)$ as shown in Figure 1a. To construct the interventional distribution, we only delete the causal edge between $X$ and $T$ and keep all the other causal relationships unchanged. In this distribution, $T$ is uniformly sampled from $\{1, 2, ..., m\}$. Note that $p^I(Y|X, T) = p^I(Y_T|X) = p^O(Y_T|X) = p^O(Y|X, T)$. As $p(Y_T|X)$ is the same for both observational distribution and interventional distribution, we omit the superscript for simplicity. Then for predicting $Y$, the objective of counterfactual maximum likelihood estimation (CMLE) would be:

$$\mathbb{E}_{p^I(X,Y,T)}\big[ -\log p_\theta^I(Y|X, T)\big] = \mathbb{E}_X\big[\frac{1}{m}\sum_{i=1}^{m}\mathbb{E}_{Y_i|X}[-\log p_\theta(Y_i|X)]\big] \tag{1}$$

Similarly, for predicting $T$, we have:

$$\mathbb{E}_{p^I(X,Y,T)}\big[ -\log p_\theta^I(T|X, Y)\big] = \mathbb{E}_X\big[\frac{1}{m}\sum_{i=1}^{m}\mathbb{E}_{Y_i|X}[-\log p_\theta^I(T = i|X, Y_T)]\big] \tag{2}$$

Here $\theta$ is the learnable parameters. In the remaining part of paper, the probabilities without superscripts are default to be observational probabilities. In the following sections, we first focus on deriving two different upper bounds of the CMLE objective for predicting $Y$: Implicit CMLE and Explicit CMLE. In Implicit CMLE, we add a regularizer to balance the distribution of $X$ given $T$ in the representation space. For Explicit CMLE, we directly generate counterfactual examples during training. To predict $T$, we train a classifier using normal MLE on a augmented dataset with the generated *counterfactual examples* to estimate the interventional distribution $p^I(X, Y, T)$. We define a counterfactual example as follows:

**Definition 2.** *A counterfactual example is an example $(x, t, y)$ with $y$ sampled from $p(Y_t|X = x)$ s.t. $(x, t, y) \notin \mathcal{D}$ and there is an example $(x, t', y') \in \mathcal{D}$ where $t' \neq t$.*

We can rewrite our objective function in Equation 1 as $\mathbb{E}_x\big[\sum_{t=1}^{m}\mathcal{L}_\theta(x, t)\big]$, by defining the expected loss function of a specific input and treatment as:

**Definition 3.** *We denote the loss function for predicting $Y$ as $L_\theta(x, t, y) = -\log p_\theta(Y_t = y | X = x)$. Then the expected loss function given $X = x$ and $T = t$ is $\mathcal{L}_\theta(x, t) = \mathbb{E}_{Y_t|X=x}[L_\theta(x, t, Y_t)]$.*

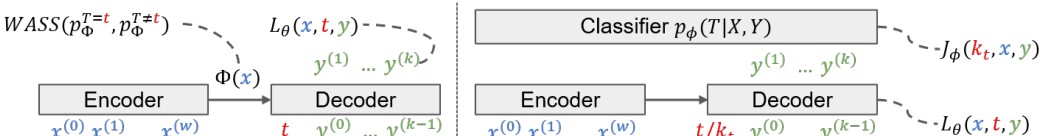

Figure 2: The model architecture of Implicit CMLE (left) and Explicit CMLE (right).

Because of the Strong Ignorability Assumption, we have $\mathcal{L}_\theta(x,t) > 0$. Then we define an *expected factual loss* $\epsilon_F^{T=t}$ and *expected counterfactual loss* $\epsilon_{CF}^{T=t}$ when observing a specific treatment $T = t$:

**Definition 4.** *The expected factual loss and expected counterfactual loss given $T = t$ are $\epsilon_F^{T=t} = \mathbb{E}_{X|T=t}[\mathcal{L}_\theta(X,t)]$ and $\epsilon_{CF}^{T=t} = \mathbb{E}_{X|T\neq t}[\mathcal{L}_\theta(X,t)]$, respectively.*

Here $\epsilon_F^{T=t}$ measures how well the model performs on the observational distribution while $\epsilon_{CF}^{T=t}$ measures how well the model would perform on an alternative world where a different treatment is chosen. Then we can rewrite our CMLE objective as:

$$\mathbb{E}_x\Big[\sum_{t=1}^m \mathcal{L}_\theta(x,t)\Big] = \sum_{t=1}^m [p(T=t)\epsilon_F^{T=t} + p(T\neq t)\epsilon_{CF}^{T=t}] \tag{3}$$

### 3.2 Implicit CMLE

We consider learning a representation function of $X$, $\Phi : \mathcal{X} \to \mathbb{R}^d$, where $\mathbb{R}^d$ is the representation space. Here we make the following assumption about $\Phi$:

**Assumption 2.** $\Phi : \mathcal{X} \to \mathbb{R}^d$ *is a invertible function. We denote $\Psi : \mathbb{R}^d \to \mathcal{X}$ as the inverse of $\Phi$. For continuous $\mathcal{X}$, $\Psi$ also needs to be differentiable to calculate the Jacobian matrix.*

Note that this assumption can be fulfilled for neural networks as long as all the activation functions are invertible. We denote the distribution induced by $\Phi$ by $p_\Phi$. We can easily transform $p(X|T)$ to $p_\Phi(\Phi(X)|T)$ by a standard change of variable [50, 51]:

**Definition 5.** *For $i \in \{1, 2, ..., m\}$ and $r \in \mathbb{R}^d$, we denote $p_\Phi^{T=i}(r) := p_\Phi(r|T=i)$. For discrete $\mathcal{X}$, we simply have $p_\Phi^{T=i}(r) = p(X = \Psi(r)|T = i)$, while for continuous $\mathcal{X}$, we have $p_\Phi^{T=i}(r) = p(X = \Psi(r)|T = i) \det\left|\frac{\partial \Psi}{\partial r}\right|$.*

We adopt a similar idea as [22] and use the Integral Probability Metric (IPM) to derive an upper bound of the CMLE objective. IPM is a class of metrics between probability distributions with the following definition [52]:

**Definition 6.** *For two probability density functions $p$ and $q$ defined over $\mathcal{S}$, we have*

$$\text{IPM}_G(p,q) := \sup_{g\in G}\left|\int_\mathcal{S} g(s)(p(s) - q(s))ds\right|$$

*Where $G$ is a family of functions $g : \mathcal{S} \to \mathbb{R}$.*

Note that here (and in the rest of our paper) we abuse the notation of integration. If $\mathcal{X}$ is discrete, the integration would become a sum over it. Then we can derive an upper bound for our CMLE objective by Assumption 2, Definition 5 and Equation 3: (See Appendix A.1 for full proof.)

**Theorem 1.** *For a function family $G$ and a constant $B_\Phi > 0$ s.t. $\frac{1}{B_\Phi}\mathcal{L}_\theta(\Psi(r),t) \in G$ for any $r \in \mathbb{R}^d$ and $t \in \{1, 2, ..., m\}$, the CMLE objective is bounded by:*

$$\mathbb{E}_x\Big[\sum_{t=1}^m \mathcal{L}_\theta(x,t)\Big] \leq \sum_{t=1}^m [\epsilon_F^{T=t} + p(T\neq t)B_\Phi \text{IPM}_G(p_\Phi^{T=t}, p_\Phi^{T\neq t})]$$

In this upper bound, we are only left with observational/factual quantities that can be estimated from the observational data. We choose $G$ to be 1-Lipschitz functions, which implies $\text{IPM}_G$ to be the Wasserstein distance [53, 54], denoted by $\text{WASS}(\cdot,\cdot)$. Then the empirical CMLE objective using an unbiased estimator of this upper bound can be written as:

$$\underset{\theta}{\arg\min} \frac{1}{n}\sum_{i=1}^{n}\frac{n}{u_{t_i}}L_\theta(x_i,t_i,y_i) + \alpha\sum_{j=1}^{m}(1-\frac{u_j}{n})\text{WASS}(\{\Phi(x_i)\}_{i:t_i=j},\{\Phi(x_i)\}_{i:t_i\neq j}) \quad (4)$$

Where $u_i = \sum_{j=1}^{n}\mathbb{1}_{t_j=i}$ and $\mathbb{1}$ is the indicator function. $\alpha$ is a hyperparameter accounting for $B_\Phi$ and we abuse the notation of $\text{WASS}(\cdot,\cdot)$ to denote an empirical version of Wasserstein distance. The resulting empirical training objective consists of a weighted version of the normal factual loss for empirical risk minimization and a regularization term to draw $p_\Phi^{T=i}$ and $p_\Phi^{T\neq i}$ closer. $\frac{n}{u_{t_i}}$ accounts for different number of examples with each label and $1 - \frac{u_j}{n}$ account for $p(T\neq j)$ (see Theorem 1). The empirical Wasserstein distance is calculated among examples with different labels in the representation space. In practice, to perform back propagation, we directly compute the gradient of $\text{WASS}(p_\Phi^{T=i},p_\Phi^{T\neq i})$ over a single mini-batch, using the algorithm proposed by [55]. (See Appendix B.1 for details.)

### 3.3 Explicit CMLE

While Implicit CMLE tries to reduce the spurious correlation by learning a better representation of $X$, we consider directly learning a better approximation of the log likelihood $\log p_\theta(Y_t|X)$ by explicitly generating counterfactual examples at training. Here we assume that there is a neural network with parameter $\phi$ that approximates the interventional distribution $p_\theta^I(T|Y,X)$ defined as follow:

**Definition 7.** $p_\phi(T|Y,X)$ *is an approximate of* $p_\theta^I(T|Y,X)$ *parameterized by* $\phi$ *s.t.* $KL(p_\theta^I(T|X,Y)||p_\phi(T|X,Y)) < B$, *where* $B > 0$ *is a real constant. We also define an associated loss function* $J_\phi(t,x,y) = -\log p_\phi(T=t|Y=y,X=x)$.

Here $KL(p||q)$ denotes the KL divergence between two distributions $p$ and $q$. Our intuition is that it is directly feasible to learn the normal MLE objective $\mathbb{E}_{X,Y,T}[\log p_\phi(T|Y,X)]$ from the observation data, while is infeasible to directly learn the CMLE objective $\mathbb{E}_x\big[\sum_{t=1}^{m}\mathcal{L}_\theta(x,t)\big]$. Also, as $T$ is a categorical variable, its distribution can be simplified as a Multinomial distribution. We then derive an upper bound of the CMLE objective using Bayes' rule and Jensen's Inequality: (See Appendix A.2 for full proof.)

**Theorem 2.** *By Assumption 1, there exist* $0 < \delta_1 < \delta_2 < 1$ *s.t.* $\delta_1 < p(T|X) < \delta_2$, *then we have*

$$\mathbb{E}_x\big[\sum_{t=1}^{m}\mathcal{L}_\theta(x,t)\big] \leq \mathbb{E}_{x,t}\big[\eta\mathbb{E}_{p_\theta(Y_t|X=x)}[L_\theta(x,t,y)] + \sum_{i\neq t}^{m}\mathbb{E}_{p_\theta(Y_i|X=x)}[J_\phi(i,x,y)]\big] + \mu$$

*Where* $\eta = (1 + \frac{1}{p(t)}(1-\frac{1}{m}))$, $\mu = (m\delta_2 - m\delta_1 - m + 1)\log m$.

This theorem indicates that we can transfer the counterfactual likelihood of predicting $Y$ into the counterfactual likelihood of $T$ predicted by another model. Note that the resulting bound would be tighter when $p_\phi(T|Y,X)$ is a better approximate of $p_\theta^I(T|Y,X)$ as we use the fact $KL(p_\theta^I(T|Y,X)||p_\phi(T|Y,X)) > 0$ in the proof (see Appendix A.2). In practice, we can either pretrain a $p_\phi(T|Y,X)$ on the observational data when $p_\theta^O(T|Y,X)$ and $p_\theta^I(T|Y,X)$ are not too far away, or we can simultaneously train a $p_\phi(T|Y,X)$ with $p_\theta(Y_i|X)$ to take advantages of the generated counterfactual examples.

Since it is expensive to take $m-1$ samples of $y_i$ from $p_\theta(Y_i|X)$ for each training example at train time, we introduce another random variable $K_t$ for each $t \in \{1,2,...,m\}$, such that is $K_t$ uniformly distributed on $\{1,2,...,m\}\backslash\{t\}$. Then the Explicit CMLE objective can be rewritten as:

$$\underset{\theta}{\arg\min} \mathbb{E}_{x,t}\Big[\mathbb{E}_{p_\theta(Y_t|X=x)}\big[L_\theta(x,t,Y_t)\big] + \frac{m-1}{\eta}\mathbb{E}_{K_t}\big[\mathbb{E}_{p_\theta(Y_{K_t}|X=x)}[J_\phi(K_t,x,Y_{K_t})]\big]\Big]$$

Here we ignore the constant term $\mu$ in the upper bound and multiply the remaining terms by $\frac{1}{\eta}$. We can then write the Explicit CMLE objective empirically as:

$$\underset{\theta}{\arg\min} \frac{1}{n}\sum_{i=1}^{n}[L_\theta(x_i,t_i,y_i) + \alpha(t_i)J_\phi(k_{t_i},x_i,y_\theta(x_i,k_{t_i}))] \quad (5)$$

Where $\alpha(t_i)$ is a hyperparameter corresponding to $\frac{m-1}{\eta}$, $k_{t_i}$ is randomly sampled from $p(K_{t_i})$ and $y_\theta(x_i, k_{t_i})$ is sampled from $p_\theta(Y_{k_{t_i}}|X = x)$. We only sample $K_t$ once and sample the corresponding $Y_{K_t}$ once for computational efficiency. In our implementation, to backpropagate through $p_\theta(Y_{k_{t_i}}|X = x)$, we adopt the Gumbel-Softmax approach [56] to deal with the discrete text data, which creates a differentiable sample to replace the non-differentiable discrete variable. For alternative approaches for discrete and continuous $\mathcal{Y}$, see Appendix B.2.

### 3.4 CMLE Implementation

While CMLE is a general causal training framework that can be applied to almost any deep learning architecture, in this paper, we primarily consider the Transformer [57] architecture, which excels at modeling sequential data and has been successfully applied to various domains, especially with pretraining on a large amount of data [58, 59]. Here we assume for any $x \in \mathcal{X}$, we can write $x$ as $x = \{x^{(0)}, x^{(1)}, ..., x^{(w)}\}$ and for any $y \in \mathcal{Y}$, we have $y = \{y^{(0)}, y^{(1)}, ..., y^{(k)}\}$. We input $T = t$ by prepend a special token/vector corresponding to $t$ as a prefix to the decoder input $y$. The model architectures of Implicit CMLE and Explicit CMLE are shown in Figure 2.

## 4 Experiments

We consider one synthetic dataset and two real-world tasks, Natural Language Inference and Image Captioning, to evaluate the effectiveness of our proposed CMLE causal training framework. Both real-world tasks have prominent datasets containing well-known spurious correlations; we aim to reduce these correlations' influence on each target task. In the following sections, we will introduce our experiments in details and compare the performance of our proposed CMLE against vanilla MLE.

### 4.1 Simulated Experiments

We consider a synthetic dataset generated by the following procedure inspired by [45]:

Given the followigng functions $g_1(x) = x - 0.5, g_2(x) = (x - 0.5)^2 + 2, g_3(x) = x^2 - 1/3, g_4(x) = -2\sin(2x), g_5(x) = e^{-x} - e^{-1} - 1, g_6(x) = e^{-x}, g_7(x) = x^2, g_8(x) = x, g_9(x) = \mathbb{1}_{x>0}, g_{10}(x) = \cos(x), g_{11}(x,t) = \log(t + x^2), g_{12}(x,t) = e^{t+x}$, and $g_{13}(x,t) = \sin(t + x)$:

1. Sample $x_i$ from $\mathcal{N}(0, 1)$, $i \in \{1, 2, ..., 100\}$.
2. Let $s = g_1(x_1) + g_2(x_2) + g_3(x_3) + g_4(x_4) + g_5(x_5)$. Let $t = 1$ if $s \leq 4$; $t = 2$ if $4 < s \leq 5$; $t = 3$ if $s > 5$.
3. Let $\mu_1 = g_6(x_1) + g_7(x_2) + g_8(x_3) + g_{11}(x_6, t) + g_{12}(x_7, t)$, $\mu_2 = g_9(x_4) + g_{10}(x_5) + g_{11}(x_8) + g_{13}(x_9, t)$. Then sample $y$ from $\mathcal{N}((\mu_1, \mu_2), 1)$.
4. Repeat steps 1-3 for N times.

Here $X$ is a real vector of length 100, $T$ takes value in $\{1, 2, 3\}$ and $Y$ is a real vector of length 2. The task is to predict $Y$ given $X$ and $T$. Then we generate 10000 train data, and 5000 validation data and 5000 in distribution (*observational*) test data by setting N=10000 and 5000 respectively. We also generate corresponding ground truth *counterfactual* test data for testing how well our models can estimate the counterfactuals. *OOD1*, *OOD2*, *OOD3* are three out-of-distribution test sets that generates $T$ using different mechanisms as follows:

- *OOD1*: $t$ is uniformly sampled from $\{1, 2, 3\}$.
- *OOD2*: Let $s = g_1(x_6) + g_2(x_7) + g_3(x_8) + g_4(x_9) + g_5(x_{10})$. Let $t = 1$ if $s \leq 4$; $t = 2$ if $4 < s \leq 5$; $t = 3$ if $s > 5$.
- *OOD3*: Let $s = g_6(x_1) + g_7(x_2) + g_8(x_3) + g_9(x_4) + g_{10}(x_5)$. Let $t = 1$ if $s \leq 4$; $t = 2$ if $4 < s \leq 5$; $t = 3$ if $s > 5$.

We use multi-layer perceptrons for all of our models and use the mean-squared-error (MSE) loss at training. This means we assume $Y$ to follow a Gaussian distribution given $X$ and $T$. In Table 1, we report MSE of predicting $Y$ on different test sets.

Here Explicit CMLE* means using a $p_\phi(T|X, Y)$ that is trained on the interventional distribution instead of the observational distribution. The numbers are averaged over ten runs and the standard deviation is also reported. We take $\alpha = 0.01$ for Implicit CMLE and $\alpha = 0.1$ for Explicit CMLE.

| test set | MLE | Implicit CMLE | Explicit CMLE | Explicit CMLE* |
|---|---|---|---|---|
| Observational | 3.63±0.20 | 3.57±0.17 | 3.45±0.29 | **3.28**±0.27 |
| Counterfactual | 5.25±0.40 | 5.11±0.38 | 4.77±0.38 | **4.61**±0.42 |
| OOD1 | 4.52±0.28 | 4.46±0.37 | 4.31±0.32 | **3.98**±0.36 |
| OOD2 | 7.25±0.56 | 7.17±0.56 | 6.94±0.73 | **6.39**±0.68 |
| OOD3 | 4.51±0.33 | 4.35±0.36 | 4.15±0.40 | **3.91**±0.44 |

Table 1: MSE of predicting $Y$ given $X$ and $T$. The numbers are averaged over ten runs.

We can see that all CMLE methods perform better than MLE on the counterfactual test set and the out-of-distribution test sets, while Explicit CMLE performs better than Implicit CMLE. And the performance of Explicit CMLE would be better when $p_\phi(T|X, Y)$ is closer to the ground truth interventional distribution.

## 4.2 Natural Language Inference (NLI)

Natural language inference (NLI) [23] is a fundamental task in natural language understanding, which aims to infer the logical relation between a pair of sentences, *premise* and *hypothesis*. Unfortunately, NLI datasets are known to present significant spurious correlations. For example, [3] demonstrated that a machine learning model can non-trivially outperform the majority baseline using only the hypothesis without the premise. One known spurious correlation comes from the background knowledge contained in the premises. i.e. The prediction is based on the background knowledge instead of the logical relation between the premise and the hypothesis. For example, the hypothesis "TV viewing and consumption both begin at about the same time." would usually be considered to be true from common knowledge. While a fine-tuned RoBERTa [60] also predict this example as entailment, it contradicts its premise "Regular TV viewing typically begins between 2 and 3 years of age, consuming about 10.". We view the premise sentence as $X$, the hypothesis sentence as $Y$, and the label (entailment, neutral, or contradiction) as $T$. The NLI annotation procedure [23, 61] naturally fits in our causal framework in Figure 1, as the crowd workers are instructed to write a hypothesis that is entailed/neutral/contradicted to a given premise.

We consider three large-scale NLI datasets: SNLI [23], MNLI [24] and ANLI [62]. SNLI and MNLI are considered standard benchmarks, with each containing 550,152/10,000/10,000 examples and 392,702/20,000/20,000 examples for train/dev/test split respectively. ANLI was constructed as a hard adversarial dataset to eliminate spurious correlations and balance the data distribution, thus it serves as a great benchmark to evaluate a model's generalizability. ANLI contains three subsets of increasing difficulty—A1, A2, and A3—with about about 392,702/20,000/20,000 examples, 45,460/1,000/1,000 examples and 100,459/1,200/1,200 examples each for train/dev/test split respectively.

For generating counterfactual hypotheses we fine-tune the pre-trained BART large model [63] provided by HuggingFace [64] with 12 encoder and decoder layers. We fine-tune BART on the combined dataset of SNLI and MNLI using both Implicit and Explicit CMLE. We choose $\alpha = 0.003$ for Implicit CMLE and $\alpha = 0.1$ for Explicit CMLE. We predict $p_\phi(T|X, Y)$ for Explicit CMLE with a conventionally fine-tuned RoBERTa large model with 24 layers. We call this model NLI-RoBERTa.

We evaluate how consistent our generated hypotheses $Y$ are to their corresponding labels $T$ using both NLI-RoBERTa and human evaluation[2] in Table 2 (right). In both cases, we perform this assessment by comparing the model- or human-predicted $t'$ against the ground truth $t$. The accuracy of the NLI-RoBERTa is shown in the first row of Table 2. Both of our CMLE methods significantly outperform the MLE baseline by a large margin on the labeling consistency of the generated hypothesis, which indicates that our CMLE framework can generate $Y_t$ that is truly influenced by $T = t$.

To construct a dataset that mimics sampling from the interventional distribution, we generate two counterfactual examples for each example in the original train dataset of SNLI and MNLI. We filter out the inconsistent examples using NLI-RoBERTa. We generate three augmented datasets using standard MLE, Implicit CMLE, and Explicit CMLE and fine-tune a copy of RoBERTa on each one. The results for these models are presented in rows *MLE (w/ Aug)*, *Implicit CMLE* and *Explicit CMLE* of Table 2 respectively. The baseline *MLE* is NLI-RoBERTa itself. We test the accuracy of all the fine-tuned RoBERTas simultaneously on the regular test sets (MNLI, SNLI) and the out-of-domain adversarial test sets (ANLI). The results in Table 2 show that, while maintaining performance on

---

[2]All of our human evaluations are conducted via Amazon Mechanical Turk. See Appendix C.3 for details.

|        | NLI Accuracy | | | | | | Generation Consistency | |
| Method | A1 | A2 | A3 | ANLI | SNLI | MNLI-m/mm | SNLI (R/H) | MNLI (R/H) |
|--------|------|------|------|------|------|-----------|-----------|-----------|
| MLE | 47.3 | 26.1 | 22.4 | 31.3 | 92.6 | 90.6/90.5 | N/A | N/A |
| MLE (w/ Aug) | 47.2 | 25.4 | 21.7 | 30.8 | 92.4 | 90.4/90.3 | 65.5/46.8 | 66.0/48.3 |
| Implicit CMLE | 47.0 | **27.1** | **24.0** | **32.2** | 92.4 | 90.4/90.5 | 88.5/69.6 | 75.1/57.7 |
| Explicit CMLE | **47.7** | 26.8 | 23.7 | **32.2** | 92.4 | 90.5/90.5 | **95.2/70.8** | **85.8/62.4** |

Table 2: Accuracy of RoBERTa fine-tuned on augmented SNLI + MNLI (left), along with label consistency (accuracy) of the generated hypothesis (right). We report all the results on test sets except on MNLI we use dev sets, where 'm' means matched and 'mm' means mismatched. 'R' indicates evaluations by NLI-RoBERTa and 'H' indicates evaluations by human.

regular test sets, our proposed CMLE methods significantly outperform both MLE baselines on the adversarial test set. This indicates that CMLE can improve the out-of-domain generalization performance of large deep learning models without hurting the in-domain performance.

## 4.3 Image Captioning

Image captioning is a fundamental task in studying multimodal interaction. However, image captioning models also suffer severely from spurious correlations [65] that lead to hallucinations. This type of spurious correlation is mainly caused by the typical co-occurrences of objects in images. For example, as toilets are frequently present in bathroom pictures, a captioning model trained by MLE is very likely to hallucinate a 'toilet' whenever it sees a 'bathroom', even if no toilet is present.

To reduce these hallucination artifacts, we apply our CMLE framework by regarding the image as $X$, the caption as $Y$, and whether the caption is hallucinated as $T$. In our setting, the non-hallucinated captions are ground truth captions written by a human, and the hallucinated captions are generated by image captioning models. The dataset we use is the MSCOCO 2014 dataset [66] which contains 123,287 images of common objects with 5 human-annotated captions per image. We use the Karpathy [67] split with 113,287/5,000/5,000 images in the train/validation/test set respectively. Our implementation is based on [68].

The Transformer architecture we use directly follows [57], with 6 layers in each of the encoder and decoder layers. To construct a dataset with both hallucinated and non-hallucinated captions, we synthetically generate captions by masking out one object word for each caption and use BERT (large) [58] to fill in an alternative word that aligns with the language model. We estimate the caption hallucination by the CHAIR score [65], which parses the object words in a caption and compares them to the ground truth objects annotated in MSCOCO images and captions. In total, we synthesize 648,550 hallucinated captions for 93,004 images in the training set.

Then we train the transformer model on this augmented dataset of MSCOCO captions with synthetic hallucinated captions using regular MLE and our proposed CMLE methods. We choose $\alpha = 0.0002$ for Implicit CMLE and $\alpha = 0.0001$ for Explicit CMLE. For Explicit CMLE, we fine-tune a pre-trained LXMERT [59] on this augmented dataset as $p_\phi(T|X, Y)$. Here we consider two MLE baselines: The *MLE* baseline is the Transformer model regularly trained on the original MSCOCO dataset, while the *MLE (w/ Aug)* baseline is trained on the augmented dataset.

We conduct user studies on the generated captions as shown in Table 3. In the user study, we let the user choose the best caption among our two baselines and two proposed CMLE methods from the following aspects: *Faithfulness*, *Expressiveness* and *Informativeness*. Faithfulness means how faithful is the generated caption to the corresponding image, which partially measures the extend of hallucinations. Expressiveness means if the generated caption is coherent, grammatically, and semantically correct. Informativeness means if the caption includes most of the information in the image instead of giving a general description. We also let the user choose the caption that best describes the image in their opinion, which is referred to as *Preference* in Table 3. We can see that our proposed CMLE methods outperform the two MLE baselines with a large margin on all the evaluation dimensions which indicates that our methods can not only reduce the hallucination contents in the generated captions but also increase the quality of generated captions in general.

The automatic evaluation results are shown in Table 4, where CHAIRs is the percentage of hallucinated sentences and CHAIRi is the percentage of hallucinated instances (objects). We can see that our CMLE methods outperform both MLE baselines on CHAIR scores, which indicates that our methods

| Choice (%) | MLE | MLE (w/ Aug) | Implicit CMLE | Explicit CMLE | Tie |
|---|---|---|---|---|---|
| Faithfulness | 20.7 | 13.3 | 28.7 | **32.0** | 5.3 |
| Expressiveness | 22.7 | 14.0 | 27.9 | **32.7** | 2.7 |
| Informativeness | 21.4 | 14.7 | 27.2 | **32.7** | 4.0 |
| Preference | 17.4 | 12.6 | 30.6 | **32.7** | 6.7 |
| Accuracy | 60.2 | 54.3 | 66.9 | **69.1** | N/A |

Table 3: Human evaluations of the quality of generated captions. We ask users to choose the best one among four generated captions according to different criterion. *Accuracy* is obtained by asking the user whether a given caption is accurate to the corresponding image or not.

| Method | BLEU@4 [69] | METEOR [70] | CIDEr [71] | SPICE [72] | CAHIRs/CHAIRi [65] |
|---|---|---|---|---|---|
| MLE | 33.1 | 26.9 | 106.7 | 20.0 | 7.3/5.3 |
| MLE (w/ Aug) | 31.3 | 26.2 | 102.1 | 19.6 | 8.0/5.8 |
| Implicit CMLE | 32.0 | 26.8 | 105.2 | 20.0 | **6.8/4.9** |
| Explicit CMLE | 32.1 | 26.6 | 105.5 | 19.9 | 7.0/5.0 |

Table 4: Automatic metrics on MSCOCO dataset for image captioning. All the results are reported on the Karpathy test split. Lower CAHIRs and CHAIRi means less object hallucinations.

can effectively reduce hallucination in image captions. At the same time, the regular scores (BLEU@4, METEOR, CIDEr, SPICE) that measure the similarities between the generated captions and ground truth captions, are comparable between CMLE and the *MLE* baseline. We note that the *MLE (w/ Aug)* baseline on augmented data is significantly worse than the *MLE* baseline on both the automatic scores and human evaluations. We conjecture that this is because the model trained by vanilla MLE is not good at finding the causal relation between the hallucination label $T$ and the caption $Y$ and thus would be confused by the mixture of hallucinated captions and non-hallucinated captions.

### 4.4 Discussions and Limitations

From both NLI and image captioning experiments, we observe that Explicit MLE is significantly better than Implicit MLE in terms of human evaluations of the generated text, and comparable on other automatic evaluations. We conjecture that this is because that Implicit CMLE only incorporates counterfactual constraints into the representation of $X$ while Explicit CMLE directly optimizes $p_\theta(Y_t|X)$ when generating the counterfactual examples.

We also note that the improvement on automatic metrics is not that large for both real-world tasks. For NLI, our data augmentation cannot perfectly simulate sampling from the interventional distribution as we only augment two counterfactual examples for each factual example. For Image captioning, as it is difficult to simulate the true distribution of hallucinated captions, the models we trained on the augmented data can be affected by these newly introduced artifacts. On the other hand, the automatic metrics like BLEU only measure the similarity of generated sentences with the observed ground truth which cannot demonstrate the full strength of our methods.

## 5 Conclusion

Our proposed CMLE framework can be applied to a wide range of learning scenarios with paired inputs and corresponding labels or involving conditional generation. While currently we only consider the spurious correlations caused by the observable confounders under the Strong Ignorability assumption, it is often the case that the confounders are not observed. For example, annotators may have a biased pattern of annotation. As we do not record the information of the annotators, this confounder is considered unobserved. Expanding the current CMLE framework by including the unobserved confounders into the causal model would be an interesting and important future direction.

## Acknowledgments

This work was supported in part by the National Science Foundation Graduate Research Fellowship under Grant No. 1650114. This material is based on work that is partially funded by an unrestricted gift from Google.

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
