# A Proofs

In this section, we give full proofs of the two main theorems in the paper.

## A.1 Implicit CMLE

Based on Definition 5, the difference between the counterfactual loss and the factual loss can be bounded by the following Lemma:

**Lemma A1.** *For a function family $G$ and a constant $B_\Phi > 0$ s.t. $\frac{1}{B_\Phi}\mathcal{L}_\theta(\Psi(r), t) \in G$ for any $r \in \mathbb{R}^d$ and $t \in \{1, 2, ..., m\}$, we have:*

$$\epsilon_{CF}^{T=t} - \epsilon_F^{T=t} \leq B_\Phi \mathrm{IPM}_G(p_\Phi^{T=i}, p_\Phi^{T \neq i})$$

*Proof.* By Definition 4, we have

$$\epsilon_{CF}^{T=t} - \epsilon_F^{T=t} = \mathbb{E}_{X|T \neq t}[\mathcal{L}_\theta(X, t)] - \mathbb{E}_{X|T=t}[\mathcal{L}_\theta(X, t)]$$

For continuous $\mathcal{X}$, we have:

$$\epsilon_{CF}^{T=t} - \epsilon_F^{T=t} = \int_\mathcal{X} \mathcal{L}_\theta(x, t)(p(X = x|T \neq t) - p(X = x|T = t))dx$$

$$= \int_{\mathbb{R}^d} \mathcal{L}_\theta(\Psi(r), t)(p(X = \Psi(r)|T \neq t) - p(X = \Psi(r)|T = t)) \det \left|\frac{\partial \Psi}{\partial r}\right| dr \quad (6)$$

$$= \int_{\mathbb{R}^d} \mathcal{L}_\theta(\Psi(r), t)(p_\Phi^{T \neq i}(r) - p_\Phi^{T=i}(r))dr \quad (7)$$

Where equality 6 is the standard change of variable and equality 7 follows from Definition 5.

For discrete $\mathcal{X}$ we have:

$$\epsilon_{CF}^{T=t} - \epsilon_F^{T=t} = \sum_\mathcal{X} \mathcal{L}_\theta(x, t)(p(X = x|T \neq t) - p(X = x|T = t))$$

$$= \int_{\mathbb{R}^d} \mathcal{L}_\theta(\Psi(r), t)(p(X = \Psi(r)|T \neq t) - p(X = \Psi(r)|T = t))dr \quad (8)$$

$$= \int_{\mathbb{R}^d} \mathcal{L}_\theta(\Psi(r), t)(p_\Phi^{T \neq i}(r) - p_\Phi^{T=i}(r))dr \quad (9)$$

Where equality 8 follows from $\Psi$'s invertibility and equality 9 follows from Definition 5. Then for both continuous and discrete $\mathcal{X}$ we have:

$$\epsilon_{CF}^{T=t} - \epsilon_F^{T=t} = B_\Phi \int_{\mathbb{R}^d} \frac{1}{B_\Phi} \mathcal{L}_\theta(\Psi(r), t)(p_\Phi^{T \neq i}(r) - p_\Phi^{T=i}(r))dr \quad (10)$$

$$\leq B_\Phi \sup_{g \in G} \left|\int_{\mathbb{R}^d} g(r)(p_\Phi^{T \neq i}(r) - p_\Phi^{T=i}(r))dr\right| \quad (11)$$

$$= B_\Phi \mathrm{IPM}_G(p_\Phi^{T=t}, p_\Phi^{T \neq t}) \quad (12)$$

Where inequality 10 follows from the definition of supremum and equality 11 follows from $\frac{1}{B_\Phi}\mathcal{L}_\theta(\Psi(r), t) \in G$ and the definition of IPM (Definition 6). $\square$

Then based on Lemma A1, we can give the following proof for Theorem 1:

*Proof.*

$$\mathbb{E}_x\big[\sum_{t=1}^{m}\mathcal{L}_\theta(x,t)\big] = \sum_{t=1}^{m}[p(T=t)\epsilon_F^{T=t} + p(T\neq t)\epsilon_{CF}^{T=t}] \tag{13}$$

$$= \sum_{t=1}^{m}[\epsilon_F^{T=t} + p(T\neq t)(\epsilon_{CF}^{T=t} - \epsilon_F^{T=t})] \tag{14}$$

$$\leq \sum_{t=1}^{m}[\epsilon_F^{T=t} + p(T\neq t)B_\Phi\text{IPM}_G(p_\Phi^{T=t}, p_\Phi^{T\neq t})] \tag{15}$$

Where equality 13 follows from Equation 3, equality 14 follow from $p(T\neq t) = 1 - p(T=t)$ and inequality 15 follows from Lemma A1 we just proved. □

### A.2 Explicit CMLE

Here we give the proof of Theorem 2:

*Proof.* We first consider expanding the counterfactual part of the CMLE objective in Equation 3 for predicting $Y$ by the Bayes rule and plug in $p_\phi(T|X,Y)$:

$$\sum_{t=1}^{m}p(T\neq t)\epsilon_{CF}^{T=t} = \sum_{t=1}^{m}p(T\neq t)\mathbb{E}_{X|T\neq t}[\mathcal{L}_\theta(X,t)]$$

$$= \sum_{t=1}^{m}p(T\neq t)\mathbb{E}_{X|T\neq t}[\mathbb{E}_{Y_t|X=x}[L_\theta(x,t,Y_t)]]$$

$$= \sum_{t=1}^{m}p(T\neq t)\mathbb{E}_{X|T\neq t}[\mathbb{E}_{Y_t|X=x}[-\log p_\theta(Y_t = y|X = x)]]$$

$$= \sum_{t=1}^{m}p(T\neq t)\mathbb{E}_{X|T\neq t}\big[\mathbb{E}_{Y_t|X}[-\log p_\phi(T=t|Y,X)$$

$$- \log p_\theta^I(T=t|X,Y) + \log p_\phi(T=t|X,Y)$$

$$- \log \sum_{i=1}^{m} p_\theta^I(Y_i|X)p_\theta^I(T=i|X) + \log p_\theta^I(T=t|X)]\big]$$

Then we let

$$\xi_0 = \sum_{t=1}^{m}p(T\neq t)\mathbb{E}_{X|T\neq t}\big[\mathbb{E}_{Y_t|X}[J_\phi(t,X,Y_t)]\big]$$

$$\xi_1 = -\sum_{t=1}^{m}p(T\neq t)\mathbb{E}_{X|T\neq t}\big[\mathbb{E}_{Y_t|X}[\log p_\theta^I(T=t|X,Y) - \log p_\phi(T=t|X,Y)]\big]$$

$$\xi_2 = \sum_{t=1}^{m}p(T\neq t)\mathbb{E}_{X|T\neq t}\big[\mathbb{E}_{Y_t|X}[-\log \sum_{i=1}^{m} p_\theta^I(Y_i|X)p_\theta^I(T=i|X) + \log p_\theta^I(T=t|X)]\big]$$

Where $J_\phi$ is defined in Definition 7. Then we can rearrange or derive an upper bound for each of these three terms as follow: (note that we abuse the integral as the summation for the discrete case)

$$\xi_0 = \sum_{t=1}^{m} p(T \neq t) \mathbb{E}_{X|T \neq t} \big[ \mathbb{E}_{Y_t|X} [J_\phi(t, X, Y_t)] \big]$$

$$= \sum_{t=1}^{m} p(T \neq t) \int_{\mathcal{X}} p(X = x | T \neq t) \big[ \mathbb{E}_{Y_t|X=x} [J_\phi(t, x, Y_t)] \big] dx$$

$$= \sum_{t=1}^{m} \int_{\mathcal{X}} p(X = x, T \neq t) \big[ \mathbb{E}_{Y_t|X=x} [J_\phi(t, x, Y_t)] \big] dx$$

$$= \sum_{t=1}^{m} \int_{\mathcal{X}} \sum_{i \neq t}^{m} p(T = i, X = x) \big[ \mathbb{E}_{Y_t|X=x} [J_\phi(t, x, Y_t)] \big] dx$$

$$= \sum_{t=1}^{m} \int_{\mathcal{X}} \Big[ \sum_{i=1}^{m} p(T = i, X = x) \big[ \mathbb{E}_{Y_t|X=x} [J_\phi(t, x, Y_t)] \big] - p(T = t, X = x) \big[ \mathbb{E}_{Y_t|X=x} [J_\phi(t, x, Y_t)] \big] \Big] dx$$

$$= \sum_{i=1}^{m} \int_{\mathcal{X}} p(T = i, X = x) \sum_{t=1}^{m} \big[ \mathbb{E}_{Y_t|X=x} [J_\phi(t, x, Y_t)] \big] dx$$

$$\quad - \sum_{t=1}^{m} \int_{\mathcal{X}} p(T = t, X = x) \big[ \mathbb{E}_{Y_t|X=x} [J_\phi(t, x, Y_t)] \big] dx$$

$$= \sum_{t=1}^{m} \int_{\mathcal{X}} p(T = t, X = x) \sum_{i=1}^{m} \big[ \mathbb{E}_{Y_i|X=x} [J_\phi(i, x, Y_i)] \big] dx$$

$$\quad - \sum_{t=1}^{m} \int_{\mathcal{X}} p(T = t, X = x) \big[ \mathbb{E}_{Y_t|X=x} [J_\phi(t, x, Y_t)] \big] dx$$

$$= \sum_{t=1}^{m} \int_{\mathcal{X}} p(T = t, X = x) \sum_{i \neq t}^{m} \big[ \mathbb{E}_{Y_i|X=x} [J_\phi(i, x, Y_i)] \big] dx$$

$$= \mathbb{E}_{x,t} \big[ \sum_{i \neq t}^{m} \mathbb{E}_{Y_i|X=x} [J_\phi(i, x, Y_i)] \big]$$

Let $z = \log p_\theta^I(T = t | X, Y) - \log p_\phi(T = t | X, Y)$, $KL(p||q)$ denotes the KL divergence between two distributions $p$ and $q$, $H(p)$ denote the entropy of a distribution $p$. Then we have:

$$\xi_1 = -\sum_{t=1}^{m} p(T \neq t)\mathbb{E}_{X|T \neq t}\big[\mathbb{E}_{Y_t|X}[z]\big]$$

$$= -\sum_{t=1}^{m} \int_{\mathcal{Y}}\int_{\mathcal{X}} z p_\theta^I(Y_t = y|X = x)p_\theta(X = x|T \neq t)p_\theta(T \neq t)dxdy$$

$$= -\sum_{t=1}^{m} \int_{\mathcal{Y}}\int_{\mathcal{X}} z p_\theta^I(T = t|X = x, Y = y)p_\theta^I(X = x, Y = y)\frac{p_\theta(T \neq t|X = x)}{p_\theta^I(T = t|X = x)}dxdy$$

$$\leq -m(1-\delta_1)\sum_{t=1}^{m} \int_{\mathcal{Y}}\int_{\mathcal{X}} \log p_\theta^I(T = t|X = x, Y = y)p_\theta^I(T = t|X = x, Y = y)p_\theta^I(X = x, Y = y)dxdy$$

$$+ m(1-\delta_2)\sum_{t=1}^{m} \int_{\mathcal{Y}}\int_{\mathcal{X}} \log p_\phi(T = t|X = x, Y = y)p_\theta^I(T = t|X = x, Y = y)p_\theta^I(X = x, Y = y)dxdy$$

$$\tag{16}$$

$$= \mathbb{E}_{p^I(X,Y)}\big[m\sum_{t=1}^{m}[-(1-\delta_1)\log p_\theta^I(T = t|X, Y) + (1-\delta_2)\log p_\phi(T = t|X, Y)]p_\theta^I(T = t|X, Y)\big]$$

$$= m\mathbb{E}_{p^I(X,Y)}\big[-(1-\delta_1)\mathbb{E}_{p_\theta^I(T|X,Y)}[\log p_\theta^I(T = t|X, Y) - \log p_\phi(T = t|X, Y)]$$

$$+ (\delta_2 - \delta_1)\mathbb{E}_{p_\theta^I(T|X,Y)}[-\log p_\theta^I(T = t|X, Y)]\big]$$

$$= m\mathbb{E}_{p^I(X,Y)}\big[-(1-\delta_1)KL(p_\theta^I(T|X, Y)||p_\phi(T|X, Y)) + (\delta_2 - \delta_1)H(p_\theta^I(T|X, Y))\big]$$

$$\leq m(\delta_2 - \delta_1)\log m \tag{17}$$

Where inequality (16) follows from $1 - \delta_2 \leq p_\theta(T \neq t|X = x) \leq 1 - \delta_1$ and the negativity of log probability. Inequality (17) follows from Definition 7 and the fact that the largest possible value of entropy of a discrete distribution with $m$ possible value is $\log m$ (can be proven by Jensen's inequality).

By Jensen's inequality, we have:

$$\xi_2 = \sum_{t=1}^{m} p(T \neq t)\mathbb{E}_{X|T \neq t}\big[\mathbb{E}_{Y_t|X}[-\log m\frac{1}{m}\sum_{i=1}^{m} p_\theta^I(Y_i|X)p_\theta^I(T = i|X) + \log p_\theta^I(T = t|X)]\big]$$

$$\leq \sum_{t=1}^{m} p(T \neq t)\mathbb{E}_{X|T \neq t}\big[\mathbb{E}_{Y_t|X}[-\frac{1}{m}\sum_{i=1}^{m}[\log p_\theta^I(Y_i|X) + \log p_\theta^I(T = i|X) + \log m] + \log p_\theta^I(T = t|X)]\big]$$

$$= \mathbb{E}_{X,Y}\big[\sum_{t=1}^{m} p(T \neq t|X, Y)[-\frac{1}{m}\sum_{i=1}^{m}[\log p_\theta^I(Y_i|X) - \log m + \log m] - \log m]\big]$$

$$= \mathbb{E}_{X,Y}\big[\sum_{t=1}^{m}[-\frac{m-1}{m}[\log p_\theta(Y_t|X)] - p(T \neq t|X, Y)\log m]\big]$$

$$= \frac{m-1}{m}\mathbb{E}_{x,t}\big[\frac{1}{p(t)}\mathbb{E}_{Y_t|X=x}[L_\theta(x, t, Y_t)]\big] - (m-1)\log m$$

Where we use the fact $p^I(T|X) = \frac{1}{m}$.

Then by adding $\xi_0$, $\xi_1$ and $\xi_2$ together, we get:

$$\sum_{t=1}^{m} p(T \neq t)\epsilon_{CF}^{T=t} = \xi_0 + \xi_1 + \xi_2$$

$$\leq \mathbb{E}_{x,t}\Big[\sum_{i\neq t}^{m} \mathbb{E}_{Y_i|X=x}[J_\phi(i,x,Y_i)]\Big] + m(\delta_2 - \delta_1)\log m$$

$$+ \frac{m-1}{m}\mathbb{E}_{x,t}\Big[\frac{1}{p(t)}\mathbb{E}_{Y_t|X=x}[L_\theta(x,t,Y_t)]\Big] - (m-1)\log m$$

$$= \mathbb{E}_{x,t}\Big[\frac{1}{p(t)}(1-\frac{1}{m})\mathbb{E}_{Y_t|X=x}[L_\theta(x,t,Y_t)] + \sum_{i\neq t}^{m}\mathbb{E}_{Y_i|X=x}[J_\phi(i,x,Y_i)]\Big] + (m\delta_2 - m\delta_1 + 1 - m)\log m$$

Then by Definition 4, the whole CMLE objective would be bounded by:

$$\mathbb{E}_x\Big[\sum_{t=1}^{m}\mathcal{L}_\theta(x,t)\Big] = \sum_{t=1}^{m}[p(T=t)\epsilon_F^{T=t} + p(T\neq t)\epsilon_{CF}^{T=t}]$$

$$= \mathbb{E}_{x,t}\Big[\mathbb{E}_{Y_t|X=x}[L_\theta(x,t,Y_t)]\Big] + \sum_{t=1}^{m}p(T\neq t)\epsilon_{CF}^{T=t}$$

$$\leq \mathbb{E}_{x,t}\Big[(1+\frac{1}{p(t)}(1-\frac{1}{m}))\mathbb{E}_{Y_t|X=x}[L_\theta(x,t,Y_t)] + \sum_{i\neq t}^{m}\mathbb{E}_{Y_t|X=x}[J_\phi(t,x,Y_t)]\Big]$$

$$+ (m\delta_2 - m\delta_1 - m + 1)\log m$$

$$\square$$

## B    Algorithm details

In this section, we give more details of the algorithms we used in the paper.

### B.1    Implicit CMLE

We adopt the same algorithm for computing the stochastic gradient of the Wasserstein distance as in [22] as shown in Algorithm 1, where $\mathrm{diag}\,(v)$ denotes the square matrix with vector $v$ as the diagonal and $\langle M, N\rangle$ denotes the dot product with two flattened matrices $M$ and $N$. More specifically, we adopt Algorithm 3 from [55].

---

**Algorithm 1:** Computing the stochastic gradient of the Wasserstein distance

---

**Input:** A random mini-batch sampled from the observation data and a representation function $\Phi_\mathbf{w}$ with current parameter $\mathbf{w}$. For each $i \in \{1, 2, ..., m\}$, there are $n_i$ examples $(x_{s_1^{(i)}}, i, y_{s_1^{(i)}}), ..., (x_{s_{n_i}^{(i)}}, i, y_{s_{n_i}^{(i)}})$ with $t = i$, and $B = \sum_{i=1}^{m} n_i$. Let $p_i = \frac{u_i}{n}$ as defined in Equation 4;

**for** $i \in \{1, 2, ..., m\}$ **do**

    Calculate the pairwise L2 distance matrix $M^{(i)}(\Phi_\mathbf{w}) \in \mathbb{R}^{n_i \times (B-n_i)}$ between all examples with $t = i$ and $t \neq i$: $M_{kl}^{(i)}(\Phi_\mathbf{w}) = \|\Phi_\mathbf{w}(x_{s_k^{(i)}}) - \Phi_\mathbf{w}(x_{s_l^{(\bar{i})}})\|$;

    Let $M = M^{(i)}(\Phi_\mathbf{w})$, $\lambda = 10/mean(M)$, $a = (p_i, ..., p_i) \in \mathbb{R}^{n_i}$ and $b = (1-p_i, ..., 1-p_i) \in \mathbb{R}^{B-n_i}$ ;

    Compute $K = \exp(-\lambda M)$ and $\tilde{K} = \mathrm{diag}\,(1/a)K$ ;

    Let $u = a$. Then **repeat**

        $u = 1/(\tilde{K}(b/K^T u))$;

    **until** *10 times*;

    Let $v = b/(K^T u)$ and calculate the approximate optimal transport matrix $T^* = \mathrm{diag}\,(u)K\,\mathrm{diag}\,(v)$ ;

    Calculate the gradient for back propagation: $g_i = \nabla_\mathbf{w}\langle T^*, M^{(i)}(\Phi_\mathbf{w})\rangle$ ;

**end**

---

In our implementation, we directly use $\langle T^*, M^{(i)}(\Phi_\mathbf{w})\rangle$ as an approximate of the $\mathrm{WASS}(\cdot, \cdot)$ term in the empirical objective function in Equation 4.

## B.2 Explicit CMLE

we separately consider the continuous case and the discrete case of $\mathcal{Y}$. For continuous $Y \in \mathcal{Y}$, we can use a reparameterization trick as proposed in [73] to separate an auxiliary noise variable $\Delta$ from $Y$ as $Y = g_\theta(X, T, \Delta)$. Then for each example, we only need to sample the noise $\Delta$ and use $g_\theta$ to get a sample of $Y$ to perform backpropagation on $\theta$.

For the discrete case, we can directly sample a $y$ from $p_\theta(Y_t | X = x)$ and then use the REIN-FORCE algorithm [74] to rewrite the gradient of the second term as $\nabla_\theta \mathbb{E}_{p_\theta(Y_i | X = x)}[J_\phi(i, x, y)] = \mathbb{E}_{p_\theta(Y_i | X = x)}[-J_\phi(i, x, y)\nabla_\theta L_\theta(x, t, y)]$. In the paper, we adopt the Gumbel-Softmax approach [56] to deal with the discrete text data, which creates a differentiable sample to replace the non-differentiable discrete variable. More specifically, we substitute a token $y$ sampled from a Multinomial distribution with $p(y = i) = \pi_i$ for $i \in \{1, 2, ..., V\}$ and $\sum_{i=1}^{V} \pi_i = 1$ by a continuous vector $z \in \mathbb{R}^V$, with:

$$z_i = \frac{\exp\left((\log \pi_i + g_i)/\tau\right)}{\sum_{j=1}^{V} \exp\left((\log \pi_j + g_j)/\tau\right)}$$

Where $g_1, g_1, ..., g_V$ are sampled i.i.d. from Gumbel(0,1). $\tau$ is the Softmax temperature controlling the sharpness of the sample $z$ across the $V$ categories. For a more detailed discussion of all three methods mentioned above, see [56].

## C Experiment Details

Our code is written with PyTorch. We use the same data split as the original dataset or as stated in Section 4 and we choose our hyperparameters by the validation performance on the dev sets.

### C.1 Natural Language Inference

**Datasets**: The SNLI dataset is licensed under a Creative Commons Attribution-ShareAlike 4.0 International License. The majority of the MNLI corpus is released under the OANC's license, and the whole MNLI dataset is in the public domain in the United States. ANLI is licensed under Creative Commons-Non Commercial 4.0.

**Training details**: We train all the models using the `Trainer` module provided by Huggingface [64] with Adam optimizer. For fine-tuning the BART-based generation models, we use a learning rate of 5e-5 and a total batch size of 128 and train for 3 epochs. Note that in our train dataset, $T$ is pretty balanced, so we treat all $\frac{n}{u_i} = 3$ and we merge this constant into the learning rate of the Implicit CMLE method (see Equation 4). For fine-tuning the RoBERTa based classification models, we adopt the same setting as [62], that is we use a learning rate of 1e-5 and a total batch size of 128 and train for 2 epochs. We conduct all our experiments on NVIDIA Titan RTX GPUs (24GB), except fine-tuning BART using the Explicit CMLE method, which is trained on NVIDIA Tesla V100 GPUs (16GB). For each experiment, we parallelize our data across four GPUs. For fine-tuning BART, MLE and Implicit CMLE takes about 13 hours to train on Titan RTX GPUs, while Explicit CMLE takes about 8 hours to train on Tesla V100 GPUs (would take a much longer time on Titan RTX GPUs). For fine-tuning RoBERTa, the augmentation methods take about 12 hours to train while the un-augmented MLE takes about 4 hours to train on NVIDIA Titan RTX GPUs.

### C.2 Image Captioning

**Dataset**: MSCOCO 2014 dataset is licensed under a Creative Commons Attribution 4.0 License.

**Training details**: We train all the models using the image captioning codebase provided by [68] with Adam optimizer. Since we train all the models from scratch, for CMLE methods, we first train the Transformer model with the normal MLE objective for the first 3 epochs and then switch to the CMLE objective to stabilize the training. For all methods, we use a learning rate of 1e-4 and a total batch size of 100 and train for 25 epochs. Note that we force the balance of $T$ among each

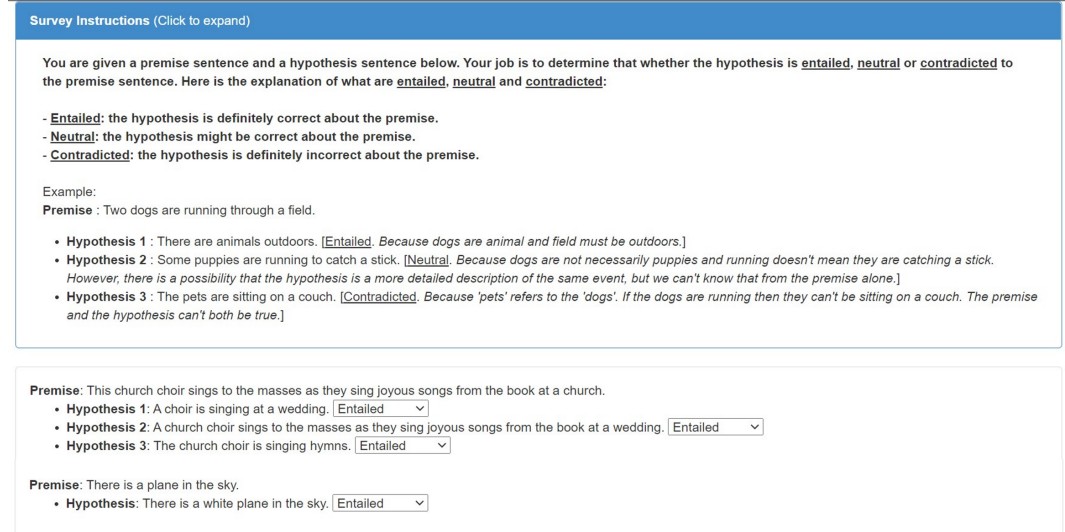

Figure 3: Natural language inference human evaluation interface.

batch, so we treat all $\frac{n}{u_i} = 2$ and we merge this constant into the learning rate of the Implicit CMLE method (see Equation 4). We conduct all our experiments on NVIDIA Titan RTX GPUs (24GB). For augmented methods, MLE and Implicit CMLE take about 40 hours to train on a single GPU while the Explicit CMLE takes about 80 hours to train on two GPUs. The un-augmented MLE takes about 20 hours to train on a single GPU.

### C.3  Mechanical Turk Details

We conduct all of our human evaluations and user studies via Amazon Mechanical Turk. Our studies only include simple multiple-choice problems that do not involve collecting any personal information. We also first manually check the examples to make sure there are no offensive contents. We are happy to provide our IRB approval document upon request.

### C.3.1  Assessing Generated NLI Hypothesis Quality

We choose 200 premises with corresponding generated hypothesis from each of the three methods, *MLE (w/ Aug)*, *Implicit CMLE* and *Explicit CMLE*, such that each hypothesis is different from each other. We produce Human Intelligence Tasks (HITs) as follows:

At the top of the HIT page, the crowdworker is shown the following instructions:

*You are given a premise sentence and a hypothesis sentence below. Your job is to determine that whether the hypothesis is entailed, neutral or contradicted to the premise sentence. Here is the explanation of what are* entailed, neutral *and* contradicted:

- Entailed: *the hypothesis is definitely correct about the premise.*
- Neutral: *the hypothesis might be correct about the premise.*
- Contradicted: *the hypothesis is definitely incorrect about the premise.*

Along with the following example and explanations:

***Premise***: *Two dogs are running through a field.*
***Hypothesis 1***: *There are animals outdoors. [*Entailed*. Because dogs are animals and the field must be outdoors.]*
***Hypothesis 2***: *Some puppies are running to catch a stick. [*Neutral*. Because dogs are not necessarily puppies and running doesn't mean they are catching a stick. However, there is a possibility that the hypothesis is a more detailed description of the same event, but we can't know that from the premise alone.]*

***Hypothesis 3****: The pets are sitting on a couch. [Contradicted. Because 'pets' refers to the 'dogs'. If the dogs are running then they can't be sitting on a couch. The premise and the hypothesis can't both be true.]*

After the instructions, the crowd worker is shown one premise with three different hypotheses and is asked to determine whether each hypothesis is entailed, neutral and contradicted to the given premise. After this question, a simple sanity checking question is also shown to check whether the worker correctly understands the instructions:

***Premise****: There is a plane in the sky.*
**Entailed Hypothesis***: There is an airplane.*
**Neutral Hypothesis***: There is a white plane in the sky.*
**Contradicted Hypothesis***: There is no plane in the sky.*

In each HIT, the same premise is shown with one of the three hypotheses at random. We only collect the example with this sanity check question answered correctly. Figure 3 is a screenshot of the user interface.

We paid 0.3 US dollar for each question and we paid $119 in total for this human evaluation. We estimate the time of answering each question would be approximately 90-120 seconds as the questions involve understanding the logical relationships between the sentences and there is a very detailed instruction to read. So the hourly wage for crowd workers would be $9 to $12.

### C.3.2   Assessing Generated Image Caption Quality

We choose 150 images with corresponding generated captions from each of the four methods, *MLE*, *MLE (w/ Aug)*, *Implicit CMLE* and *Explicit CMLE*, such that each caption is different from each other. We produce Human Intelligence Tasks (HITs) for as follows:

At the top of the HIT page, the crowd worker is shown the source image, with the instructions "*Look at the image, then read the following 4 **descriptions**. For each, answer if it is an **accurate** description (is it a fair description of the image?) Then, answer 4 **comparison questions**, choosing which sentence is the most correct, has the best style, describes the most information, and is the best overall.*"

In the **descriptions** section of the HIT page, the crowd worker then sees the four candidate description sentences (in random order) in the left-hand column. Next to each sentence, in a right-hand column, is the question "*Is this description accurate?*" and a radio button with which the crowd worker can answer "*yes*" or "*no*".

They are then shown a set of four **comparison questions** to rank the captions in terms of *correctness*, *expressiveness*, *informativeness*, and general *preference*. The questions are:

**Correctness:** *Which description is the most correct (does not describe anything that is not present in the image)?*

**Expressiveness:** *Which sentence has the most correct grammar and best style?*

**Correctness:** *Which description accurately describes the most information in the image?*

**Correctness:** *Overall, which caption would you choose to best describe the image?*

For each comparison question, the crowdworkers choose between five multiple-choice answers: *Description 1*, *Description 2*, *Description 3*, *Description 4*, or *None of them*.

Each HIT is shown to a total of 7 crowd workers. To produce a human assessment of the **accuracy** of a given caption generation technique, we simply count the rate at which all respondents answer "*yes*" to the "*Is this description accurate*" questions for each model's generated examples. To assess the **faithfulness**, **expressiveness**, **informativeness**, and **preference**, we choose the "best" generated caption for each comparison using the majority vote of the 7 crowd workers. If the majority choose "none of them" or there is a tie, we assign that image as a tie. We then produce each model's comparison question scores by counting the rates at which their captions were rated "best" for each question. Figure 4 is a screenshot of the user interface.

Figure 4: Image captioning user study interface.

We paid 0.1 US dollar for each question and we paid $126 in total for this user study. We estimate the time of answering each question would be approximately 30-40 seconds as the questions do not involve heavy reading or reasoning. So the hourly wage for crowd workers would be $9 to $12.

## D Broader Impact

Techniques for provably reducing the influence of spurious correlations on classification decisions and generative system outputs could be a boon for the reliability of and trust in general public-facing AI systems. Considering that these spurious correlations driven by observable confounders are present across a wide array of datasets and tasks, the techniques we propose herein could be broadly applicable for general system reliability improvements in supervised learning settings.

As for risks, we estimate that our proposed methods are not particularly rife for abuse, and pose a low level of danger broadly comparable to other innovations in optimizers, loss functions, and neural network architectures. We believe there is not a significant risk of any code or data produced for this study being used to nefarious ends. A possible risk to the environment is induced by the use of large pre-trained models, which would consume a large amount of computing resources at training. This issue can be mitigated by adopting more efficient training algorithms and compress the number of trainable parameters with a small trade-off in performance.