# OpenReview forum: "Counterfactual Maximum Likelihood Estimation for Training Deep Networks"
_NeurIPS.cc/2021/Conference — NeurIPS 2021 Poster_

### Official Review · Reviewer_715u · 2021-07-11

**Rating:** 7
**Confidence:** 2

**Summary:**

This paper proposed to use counterfactual maximum likelihood estimation to learn deep learning models that are less susceptible to spurious correlation relationships. In particular, it proposes two general algorithms, Implicit CMLE and Explicit CMLE, for learning causal predictions of DL models under observational data. The proposed method shows improvement over the regular MLE method in two real data sets.

**Limitations And Societal Impact:**

Yes

**Main Review:**

Overall, I think this is a high quality paper with solid works done. Let me summarize my comments in some different aspects:

1. Originality: the proposed algorithm is novel and the idea itself is very interesting.

2. Clarity: this work is well-presented in general. With intuitive explanation following each formula/theorem.

3. Significance: The method itself is very general. The two real data experiments are very enlightening and I think the proposed method has great application potential in more real life tasks.

**Time Spent Reviewing:**

4 hours

---

> ### Author Response · Authors · 2021-08-09
> **Response**
>
> Thank you for your positive review!

---

### Official Review · Reviewer_i8Y6 · 2021-07-16

**Rating:** 7
**Confidence:** 4

**Summary:**

In this work, authors propose the Counterfactual Maximum Likelihood Objective CMLE, where the parameters of the statistical model are inferred by maximizing the likelihood of the data under the interventional distribution, as opposed to the observational data. For the assumed model X -> Y <- T and X -> T, where X is the covariate, T is the treatment and Y is the outcome, the objective is to learn the parameters \theta that predict Y|X=x,T=t using just the causal link T->Y and not the spurious features in X=x. Ideally, if one could intervene on T and set do(T=t), it is possible to collect data where the outcome is observed upon the intervention done on T. Then maximizing the following likelihood: (CMLE objective) \argmax_\theta \sum_t E_{X|t}  E_{Y|x,t} p_\theta(Y|X=x,do(T=t)) would give us the MLE estimate of the parameter \theta under the set of interventional distributions do(T=t), thereby eliminating any effect of the spurious features in X on the prediction of the outcome label Y. The paper proposes two upper bounds for the above CMLE objective: i) Implicit MLE: focuses on obtaining better representations of the covariates X, such that the distribution of these representations is similar (by Wasserstein distance) for every value of T; and ii) Explicit MLE: where counterfactual examples are generated during training. Expectations in the theoretical derivations of the above upper bounds, can be estimated by Monte Carlo evaluations using only the observational data. Results are provided on two downstream tasks where models often rely upon spurious correlations: i) Natural Language Inference (NLI); and ii) Image Captioning (IC).

**Limitations And Societal Impact:**

Most of the concerns I laid above could be addressed in an expanded limitations section. I think if you condense your introduction section, you'll have a bit more space to expand your discussion. Parts of your broader impact discussion in the Appendix can also be brought here once you have more space.

**Main Review:**

- The empirical results on NLI and image captioning seem fairly convincing, especially the human evaluations. But the improvements in automated metrics don’t seem very significant.
- The paper is mostly clear and easy to understand. But, I would recommend the authors to improve the notation schema for Explicit MLE while providing an intuitive explanation for the upper bound.
- Since Explicit MLE does much better on human evaluation for both NLI and IC, it is unclear if practitioners should use Explicit MLE over Implicit MLE for all downstream tasks which require OOD generalization. It would be useful to be aware of at least one setting where Implicit MLE is useful. Although the authors conjecture a plausible explanation, the true reasoning behind this phenomena is not investigated in this work. Undoubtedly, it would be useful for the community to have a better understanding of why modeling the representations of X to be invariant under different treatments is not helpful---especially since multiple works have been proposed along these lines. Is the reason for this purely statistical, wherein the computation of the Wasserstein distance in high dimensions is inefficient or is it more fundamental, i.e. does the upper bound not closely track the CMLE objective to begin with?
- The theoretical derivations for the upper bounds seem correct. But, both upper bounds are in expectation, i.e. the authors don’t provide probabilistic tail bounds. This is especially important since even though the final optimization objective is an unbiased estimate of the upper bound, the statistical error is very high for those in high dimensions.
- The authors have provided full access to the code for both NLI and IC tasks. The implementation details have also been made fairly clear.

**Time Spent Reviewing:**

9

---

> ### Author Response · Authors · 2021-08-09
> **Response**
>
> Thank you for your positive review.
>
> For the NLI experiment, we want to point out that while the improvement of accuracy on ANLI is small, we keep almost the same performance as MLE on standard MNLI and SNLI test sets, which is very close to the current state-of-the-art performance.
>
> For the image captioning experiment, the automatic metrics themselves are not perfect. The CHAIR scores only compare a generated caption with an incomplete list of objects in the corresponding image, and the regular BLEU score only reflects the similarity between a generated caption and the ground truth captions. Thus human evaluation would be more comprehensive and accurate than automatic metrics.
>
> About the comparison between Implicit CMLE and Explicit CMLE, we conjecture that Implicit CMLE is a looser bound than Explicit CMLE, as in Implicit CMLE, the CMLE objective is bounded by a supremum. We will give a closer analysis of this and the probabilistic tail bounds.
>
> Thank you for the suggestion of expanding the limitation section. We will address the issues you mentioned in the revision.

---

> > ### Comment · Reviewer_i8Y6 · 2021-08-24
> > **Thank you**
> >
> > I appreciate your response. It would be great if you could update the draft to expand discussion on these issues. I'm inclined to keep my score the same.

---

### Official Review · Reviewer_PCZg · 2021-07-17

**Rating:** 6
**Confidence:** 2

**Summary:**

The paper tackles the problem of spurious correlations caused by observed confounders and offers two potential solutions: implicit and explicit counterfactual maximum likelihood estimation. Specifically the paper deals with the setup of predicting the outcome Y of some action T on X and considers X to potentially be a confounder. The paper compares the performance of the proposed methods to on a natural language inference and an image captioning task and shows improvements in terms of human evaluation but little difference in terms of automatic evaluation.

**Limitations And Societal Impact:**

See other comments.

**Main Review:**

Pros:
- The paper tackles the important problem of deep learning models picking up spurious correlations and proposes an interesting approach to dealing with this problem.
- The causal angle to reducing spurious correlations is currently being studied from various angles and seems to be a promising direction to handle this problem.

Cons:
- The paper assumes full observability of the confounder, that the exogenous variables are observed and only models a very specific causal model with three variables. This greatly limits the applicability of this work to the general problem of reducing reliance on spurious correlations.
- The general problem setup of the causal relations and the observability of exogenous variables seems like it lacks some clarity.
- The main benefits of the method seem to only be observable under the setting of human evaluation. It would be useful to turn to more standard benchmarks or at least similar set ups that test domain generalisation to other types of spurious correlation. The usual benchmarks such as coloured MNIST or ideas from e.g. [2] do not fit the assumed causal model though.

Detailed comments:
- Line 61: differnce -> difference
- The paper assumes the exogenous variable to be included in X - as such it can be argued that the paper does not actually consider counterfactuals but rather only interventions. Throughout the paper the definition of counterfactuals is not precise.
- Section 3.2 mentions that neural networks are invertible as long as the activations functions are invertible. This is not exactly true as not every weight matrix is invertible, e.g. check out [1] for more on this.
- Section 4.1 mentions that CMLE significantly outperforms the other baselines the adversarial test set. This is not possible to judge without any confidence intervals.
- The evaluation of the MLE with augmentation in section 4.2. seems flawed as the augmentations introduce noisy data which can be assumed to deteriorate the performance.
- The implicit CMLE seems similar to ideas in [2]. Is this correct? It would be good to discuss the relationship to this work.

[1] Kingma, Diederik P., and Prafulla Dhariwal. "Glow: Generative flow with invertible 1x1 convolutions." arXiv preprint arXiv:1807.03039 (2018).
[2] Makar, Maggie, et al. "Causally-motivated Shortcut Removal Using Auxiliary Labels." arXiv preprint arXiv:2105.06422 (2021).

**Time Spent Reviewing:**

3

---

> ### Author Response · Authors · 2021-08-09
> **Response**
>
> Thank you for your review. We would like to clarify several points in our paper as below:
>
> The full observability of the confounder $X$, comes from a commonly used assumption in the treatment effect estimation literature: the Strong Ignorability Condition (see line 152 and [3]-[8]). The aim of this assumption is to transfer the interventional distribution to observational distribution (see line 153). We are aware there could be other unobserved confounders (see line 370). While we focus on the fully observed case in this paper, we will investigate the unobserved case in the future.
>
> About the definition of counterfactuals, we do not calculate the counterfactuals by the usual $p(Y_t = y’ | T \neq t, Y = y)$. In this paper, we use counterfactual as an adjective and define the counterfactual loss by taking expectation w.r.t. $p(X|T \neq t)$ of the expected log likelihood loss function (see line 180). Shalit et. al. used a similar definition in their paper [3].
>
> About the unclear description of the exogenous variable, thank you for pointing this out and we would like to restate our setting in the following way: instead of performing MLE on the interventional distribution shown in Figure 1(c), we perform MLE on the imaginary distribution shown in Figure 1(d). That is, while keeping all the other causal relationships unchanged, we delete the causal edge between $X$ and $T$. And in this distribution, $T$ is uniformly sampled from $\{1,2,...,m\}$. Then Equation (1) would be rewritten as:
> $$
> E_{X,Y,T}\big[ -\log{p_\theta(Y_T|X)} \big] = E_X \big[ \frac{1}{m}\sum_{i=1}^m E_{Y_i|X} [-\log{p_\theta(Y_i|X)}] \big]
> $$
>
> In this way we recover our training objective. The exogenous variable of $Y$ is not contained in $X$ thus is not observed and each $Y_i$ has an independent choice of its exogenous variable. We will clarify this in the revision.
>
> About the possible applications of our proposed three variable causal model, there are mainly two broad kinds of tasks: relationship prediction and conditional generation. These tasks are common in both vision and language applications (see line 66-75). Note that we are not trying to solve every kind of machine learning problem by our proposed model.
>
> About the invertibility of neural networks, we agree that both the activation function and the weight matrix need to be invertible. Thank you for pointing this out.
>
> [2] is indeed a very relevant and interesting new work. Thank you for pointing this out. We will include it in our related work.
>
> About the confidence interval of our experiments, we did not perform multiple runs because our experiments are computationally expensive while we have limited computational resources (see appendix line 649-673). And for evaluation on standard benchmarks, as you pointed out, there is no suitable usual benchmark dataset for our model. So in the general response, we provide a new experiment on a synthetic dataset with different spurious correlations in multiple test sets.
>
> For the NLI experiment, we want to point out that while the improvement of accuracy on ANLI is small, we keep almost the same performance as MLE on standard MNLI and SNLI test sets, which is very close to the current state-of-the-art performance.
>
> For the image captioning experiment, the automatic metrics themselves are not perfect. The CHAIR scores only compare a generated caption with an incomplete list of objects in the corresponding image, and the regular BLEU score only reflects the similarity between a generated caption and the ground truth captions. Thus human evaluation would be more comprehensive and accurate than the automatic metrics.
>
> 3.  Uri Shalit, Fredrik D Johansson, and David Sontag. Estimating individual treatment effect: generalization bounds and algorithms. In Proceedings of the 34th International Conference on Machine Learning-Volume 70, pp. 3076–3085. JMLR. org, 2017.
>
> 4.  Sheng Li and Yun Fu. Matching on balanced nonlinear representations for treatment effects estimation. In Advances in Neural Information Processing Systems, pp. 929–939, 2017.
>
> 5.  Fredrik Johansson, Uri Shalit, and David Sontag. Learning representations for counterfactual inference. In International conference on machine learning, pp. 3020–3029, 2016.
>
> 6.  Ahmed M Alaa and Mihaela van der Schaar. Bayesian inference of individualized treatment effects using multi-task gaussian processes. In Advances in Neural Information Processing Systems, pp. 3424–3432, 2017.
>
> 7.  Jinsung Yoon, James Jordon, and Mihaela van der Schaar. Ganite: Estimation of individualized treatment effects using generative adversarial nets. International Conference on Learning Representations (ICLR), 2018.
>
> 8.  Liuyi Yao, Sheng Li, Yaliang Li, Mengdi Huai, Jing Gao, and Aidong Zhang. Representation learning for treatment effect estimation from observational data. In Advances in Neural Information Processing Systems, pp. 2633–2643, 2018.

---

> > ### Comment · Reviewer_PCZg · 2021-09-02
> > **Follow-Up**
> >
> > Thanks for the detailed response to my comments! Reading through your comments as well as the other reviews I decided to update my score to a 6.

---

### Official Review · Reviewer_XKW8 · 2021-07-17

**Rating:** 6
**Confidence:** 4

**Summary:**

This paper applies methodology from CATE estimation to do maximum likelihood with respect to samples from the interventional distributions Y|do(T),X, which they call Counterfactual MLE (CMLE). They provide two algorithms, Implicit CMLE and Explicit CMLE. The implicit variant uses the CATE generalization bounds from Shalit's CATE generalization bounds paper to get bounds on the likelihood from the desired counterfactual sampling distribution. The Explicit variant writes bounds in terms of the distribution T|Y,X. The two methods are applied to natural language inference and image captioning tasks. The authors suggest that this methodology and the surrounding literature on causally-inspired prediction should play a large role in the standard approach of deep learning in order to avoid shortcut learning / relying on spurious correlations.

**Ethical Concerns:**

None.

**Limitations And Societal Impact:**

Yes

**Main Review:**

1. In the intro, it is not clear why you introduce a graph that does not have an arrow from X to T but you call X a confounder. I will ask a series of subquestions on this topic:

- I understand that in general you are concerned with some spurious correlations. However, the *Precise* context in which the red X->T arrow appears was not totally clear in the paper. Y is a common child of X,T so conditioning on Y correlates X and T. But when do we condition on Y? We would usually not do this when predicting Y.

- Separately, you mentioned X may contain some confounders (mutual parent of T,Y). Wouldn't that necessitate a regular (black) arrow from X to T? Why is that not in the graphical models?

- Finally, usually spurious correlations are defined as a correlation between the outcome and some variable that we do not want to predict with. This can arise when they have a common child that is conditioned on, or a common parent that is unobserved. Then, people worry about spurious correlations because the correlation may not hold in new populations / test sets. Can you elaborate on the precise meaning of spurious correlation, conditioning on which variables makes it happen, why do you assume it is bad if a model takes advantage of the correlation (eg some assumption about this correlation not holding in the test data)


2. Why do you call ptheta(T|X,Y) in eq. (2) the "interventional distribution" and call it the "observational distribution" in line 223? By interventional do you mean "distribution over possible interventions (treatments)" and likewise by observational do you mean "distribution that determines which Yt we observe"? I would maybe rephrase both of these since "interventional distribution" is usually used for p(outcome|do(treatment))"

3. I would suggest omitting some vague descriptions/adjectives/adverbs such a "much easier" in 228 and "potentially complicated" in 229 or rephrase with more precise terms.

4. Implicit CMLE: it's pretty unclear what is the contribution from the Shalit generalization bounds paper versus what you do on top of it. Can you try to restructure so that you can make that more clear? Do I understand that the main step here is to bound the MLE objective where expectation is over interventional distribution p(Y|do(T),X), and do I understand this bound is achieved by re-arranging the terms in the Shalit generalization bound?

5. Explicit CMLE: when you assume you can estimate  P(T|Y,X) can you provide more info on what making this assumption implies? And if there is any insight on what to expect when it is misestimated?

5. M-1 samples versus tuning gumbel-softmax's temperature parameter. I'm really not sure which is worse. I suggest a dedicated synthetic experiment where you compare these two methods (does gumbel-softmax work well enough).

6. More generally, I think the omission of synthetic experiments is one of the main weaknesses of the paper. The setup has so many moving parts (estimating IPMs, learning auxiliary models, gumbel-softmax). It's so important to isolate the effects of any of these steps and the effect on the entire objective/procedure when any one of these step is wrong. For example, you could try a controlled simulation where the p(T|Y,X) model perfectly matches the ground truth.

side note not factoring into my review: for unobserved confounders, I would check out the recent work on proxy variables. There they discuss observed children of confounders and it may be a useful basis for extensions.

To summarize, these should be improved:
- writing clarity about X->T edge
- writing clarity about contributions over Shalit bounds (e.g. we re-arrange their equation (XYZ) to get our .....)
- synthetic experiment should be added to test individual parts of the setup.


**Time Spent Reviewing:**

5

---

> ### Author Response · Authors · 2021-08-09
> **Response**
>
> Thank you for your detailed review. We would like to clarify several points in our paper as below:
>
> About the X->T edge, it is indeed a regular edge as every other back arrow. We color it red just to highlight that it is the unwanted edge in our setting. We assume this red edge appears in the observed distribution (i.e. train data) as shown in Figure 1(a). In the test domain, the X->T edge may or may not exist, and the meaning/function associated with this edge may change. (we show this in the simulated experiment in the general response) So in our setting, X->T causes an unwanted spurious correlation between Y and T as this correlation would change in the test domains. Figure 1(d) is a different imaginary distribution that keeps all the other causal relationships unchanged, except deleting the causal edge between X and T. And in this distribution, T is uniformly sampled from 1,2,...,m. We want to perform MLE on this imaginary distribution instead of the observed distribution.
>
> About the interventional distribution, we use it to refer to the imaginary distribution in Figure 1(d). And the $p_\theta(T|X,Y)$ in line 223 should not have $\theta$. Sorry for the typo. We will rephrase the “interventional distribution” and other vague descriptions in our paper.
>
> About our contributions over the Shalit bound, we use and extend their Lemma 1 to the categorical case as the treatment variable T is binary in Shalit’s paper. On the other hand, our Implicit CMLE has a different meaning from the Shalit generalization bound: Implicit CMLE is an upper bound of the expected log likelihood over the interventional/imaginary distribution, while Shalit bound is an upper bound of the expected Precision in Estimation of Heterogeneous Effect (PEHE) over the observed distribution.
>
> About the estimation of $p(T|X,Y)$, we assume we can learn a predictor $p_\phi(T|X,Y)$ on the observation data. The generalization bound of this predictor is bounded by the empirical error and model complexity terms [1]. The closer this $p_\phi(T|X,Y)$ is to the interventional / imaginary distribution $p_\theta(T|X,Y)$, the tighter the Explicit CMLE upper bound would be. This is reflected in the derivation of the Explicit CMLE bound. In the new simulated experiment, we train a predictor directly on the interventional/imaginary distribution, which results in better performance of the Explicit CMLE method.
>
> We did not include the comparison between gumbel-softmax and REINFORCE because we consider this as a design choice that does not contribute to our main idea. We indeed tested both gumbel-softmax and REINFORCE in our experiments, and we did not observe significant differences between them. For a more detailed comparison between these two methods, please refer to [2].
>
> 1.  Shalev-Shwartz, Shai and Ben-David, Shai. Understanding machine learning: From theory to algorithms. Cambridge UnIversity Press, 2014.
>
> 2.  Jang, E., S. Gu, B. Poole. Categorical reparameterization with gumbel-softmax. arXiv preprint arXiv:1611.01144, 2016.

---

### Official Review · Reviewer_S8oA · 2021-07-17

**Rating:** 4
**Confidence:** 4

**Summary:**

This work proposes counterfactual maximum likelihood estimation (CMLE), a new training objective for suppressing the spurious effect of confounding variables. The authors derive two types of upperbound formulation for interventional log-likelihood (i.e. implicit CMLE and explicit CMLE). Using natural language inference (NLI) and image captioning as the intervention modeling tasks, the proposed method demonstrated superior performance in terms of human-perceived performance.

**Main Review:**

Pros
- The proposed method performed superior performance in terms of human evaluation for NLI and image captioning, compared to simple MLE baselines.

Cons
- In contrast to the effort that went into deriving the CMLE objective, the experiments seem to lack some important factors as below.
- Lack of baseline models: the aim of this paper is counterfactual estimation (both NLI and image captioning experiments were set up as counterfactual estimation tasks). However, the authors fail to include any existing works related to counterfactual estimation (or treatment effect estimation) such as [1-8].
- Strange choice of datasets: Most existing counterfactual estimation studies use both simulated datasets and real-world dataset. The former is especially used because simulated datasets contain both factual and counterfactual samples, therefore the model can be tested to see if it actually learned the intervention effect (i.e. treatment effect) properly. However, in this work, the authors employ rather unconventional NLP datasets to test their method. If the authors would like to claim their work as a counterfactual estimation model, they should test their model on simulated datasets as well, rather than only relying on human perception of how appealing the generated outputs are.

References
1. Fredrik Johansson, Uri Shalit, and David Sontag. Learning representations for counterfactual inference. In International conference on machine learning, pp. 3020–3029, 2016.
2. Uri Shalit, Fredrik D Johansson, and David Sontag. Estimating individual treatment effect: general- ization bounds and algorithms. In Proceedings of the 34th International Conference on Machine Learning-Volume 70, pp. 3076–3085. JMLR. org, 2017.
3. Ahmed M Alaa and Mihaela van der Schaar. Bayesian inference of individualized treatment effects using multi-task gaussian processes. In Advances in Neural Information Processing Systems, pp. 3424–3432, 2017.
4. Sheng Li and Yun Fu. Matching on balanced nonlinear representations for treatment effects estimation. In Advances in Neural Information Processing Systems, pp. 929–939, 2017.
5. Jinsung Yoon, James Jordon, and Mihaela van der Schaar. Ganite: Estimation of individualized treatment effects using generative adversarial nets. International Conference on Learning Repre- sentations (ICLR), 2018.
6. Liuyi Yao, Sheng Li, Yaliang Li, Mengdi Huai, Jing Gao, and Aidong Zhang. Representation learning for treatment effect estimation from observational data. In Advances in Neural Information Processing Systems, pp. 2633–2643, 2018.
7. Bryan Lim, Ahmed Alaa, and Mihaela van der Schaar. Forecasting treatment responses over time using recurrent marginal structural networks. In Advances in Neural Information Processing Systems, pp. 7493–7503, 2018.
8. Bica, I., Alaa, A.M., Jordon, J. and van der Schaar, M., 2020, September. Estimating counterfactual treatment outcomes over time through adversarially balanced representations. In International Conference on Learning Representations.

**Time Spent Reviewing:**

2.5

---

> ### Author Response · Authors · 2021-08-09
> **Response**
>
> Thank you for your review. We would like to clarify an important **misunderstanding** that the goal of this paper is not proposing a new counterfactual estimation model. Instead, we want to use counterfactual estimation models to mitigate an important problem in deep learning: that is, the learning of spurious correlations. So we did not evaluate our model on conventional observational study datasets, nor do we compare our model with other counterfactual estimation models. To the best of our knowledge, we are among the first to frame the spurious correlation problem in a similar setting to the treatment effect estimation problem. We propose two methods that can be applied to large scale deep learning models with theoretical bounds. And we prove the effectiveness of our methods by applying to two well-known real-world tasks that suffer from the spurious correlation problem.
>
> We thank you for bringing up the papers that potentially could serve as our baselines. However, as the focus of this work is on improving large scale deep learning models like pretrained transformers, some of them are not suitable for an adaptation to our problem (e.g. [3], [4], [7], [8]). On the other hand, there are some papers (e.g. [1], [2], [5], [6]) that can possibly be adapted to our problem. The Implicit CMLE method we propose is in fact an adaptation of [2]. Our core goal is to demonstrate that causality-inspired approaches can reduce the spurious correlation problem, a novel finding. The scope of this work is not to rigorously evaluate every possible treatment effect or counterfactual estimation approach head-to-head, and we leave investigating them to future work.
>
> About simulated dataset, we add a new simulated experiment in the general response and show the effectiveness of CMLE on the counterfactual test set and several out-of-distribution test sets.

---

### Author Response · Authors · 2021-08-09
**General response**

We thank all reviewers for their insightful comments. We add a new experiment on a synthetic dataset generated by the following procedure:

Given $g_1(x) = x−0.5$, $g_2(x) = (x − 0.5)^2 + 2$, $g_3(x) = x^2 − 1/3$, $g_4(x) = −2 \sin(2x)$, $g_5(x) = e^{−x} − e^{−1} − 1$, $g_6(x) = e^{−x}$, $g_7(x) = x^2$, $g_8(x) = x$, $g_9(x) = 1_{x>0}$, $g_{10}(x) = \cos(x)$, $g_{11}(x,t) = \log(t + x^2)$, $g_{12}(x,t) = e^{t + x}$, and $g_{13}(x,t) = \sin(t + x)$,

1. Sample $x_i$ from $\mathcal{N}(0,1)$, $i \in \{1,2,...,100\}$.

2. Let $s = g_1(x_1) + g_2(x_2) + g_3(x_3) + g_4(x_4) + g_5(x_5)$.

Then $t = 0$ if $s \leq 4$; $t = 1$ if $4 < s \leq 5$; $t = 2$ if $s > 5$.

3. Let $\mu_1 = g_6(x_1) + g_7(x_2) + g_8(x_3) + g_{11}(x_6, t) + g_{12}(x_7, t)$, $\mu_2 = g_9(x_4) + g_{10}(x_5) + g_{11}(x_8) + g_{13}(x_9, t)$. Then sample $y$ from $\mathcal{N}((\mu_1, \mu_2), 1)$.

4. Repeat steps 1-3 for N times.

Our data generation procedure is inspired by [1], which uses a similar set of formulas to construct the causal relation between $X$, $Y$ and $T$ that fit in our assumed causal structural model. We make the causal relationships slightly more complicated than [1] by making $T$ a categorical variable instead of binary and adding functions $g_{11}$, $g_{12}$, $g_{13}$.

Here $X$ is a real vector of length 100, $T$ takes 0 or 1 or 2 and $Y$ is a real vector of length 2. The task is to predict $Y$ given $X$ and $T$. This data generation procedure is inspired by [1]. Then we generate 10000 train data, and 5000 validation data and 5000 test data by setting N=10000 and 5000. We also generate corresponding ground truth counterfactual test data for testing how well our models can estimate the counterfactuals. OOD1, OOD2, OOD3 are three out-of-distribution test sets that generates $T$ using different mechanisms as follows:

OOD1: $t$ is uniformly sampled from $\{1,2,3\}$.

OOD2: Let $s = g_1(x_6) + g_2(x_7) + g_3(x_8) + g_4(x_9) + g_5(x_{10})$.

Then $t = 0$ if $s \leq 4$; $t = 1$ if $4 < s \leq 5$; $t = 2$ if $s > 5$.

OOD3: Let $s = g_6(x_1) + g_7(x_2) + g_8(x_3) + g_9(x_4) + g_{10}(x_5)$.

Then $t = 0$ if $s \leq 4$; $t = 1$ if $4 < s \leq 5$; $t = 2$ if $s > 5$.

We use multi-layer perceptrons for all of our models and use the mean-squared-error (MSE) loss at training. In the table below, we report MSE of $y$ on different test sets. The numbers are averaged over ten runs and the standard deviation is also reported.

|	|MLE	|Implicit CMLE	|Explicit CMLE	|Explicit CMLE*	|
|---| ----- | ---- | ---- | --- |
|Observational	|3.63±0.20	|3.57±0.17	|3.45±0.29	|**3.28**±0.27	|
|Counterfactual	|5.25±0.40	|5.11±0.38	|4.77±0.38	|**4.61**±0.42	|
|OOD1	|4.52±0.28	|4.46±0.37	|4.31±0.32	|**3.98**±0.36	|
|OOD2	|7.25±0.56	|7.17±0.56	|6.94±0.73	|**6.39**±0.68	|
|OOD3	|4.51±0.33	|4.35±0.36	|4.15±0.40	|**3.91**±0.44	|

Here Explicit CMLE* means using a $p_\phi(T|X,Y)$ that is trained on the interventional distribution instead of the observational distribution. The numbers are averaged over ten runs and the standard deviation is also reported. We take $\alpha=0.01$ for Implicit CMLE and $\alpha=0.1$ for Explicit CMLE.

We can see that all CMLE methods perform better than MLE on the counterfactual test set and the out-of-distribution test sets, while Explicit CMLE performs better than Implicit CMLE. And the performance of Explicit CMLE would be better when $p_\phi(T|X,Y)$ is closer to the ground truth interventional distribution.

1.  Sheng Li and Yun Fu. Matching on balanced nonlinear representations for treatment effects estimation. In Advances in Neural Information Processing Systems, pp. 929–939, 2017.

---

### Decision · Program_Chairs · 2021-09-27

**Decision:**

Accept (Poster)

**Comment:**

We thank the authors for their engaging submission on an important topic of research.

A major shared concern of the reviewers is the lack of baseline methods in the empirical study.  While the related work section describes the state of existing invariant learning algorithms, the authors make no direct comparisons in the experiments.  And while there may be no directly comparable algorithm, the wealth of causal representation learning algorithms enumerated by reviewer S8oA could still be used to form predictions that highlight the key differences of the CMLE framework.

Still though, this work introduces and empirically investigates a new objective for learning invariant representations.  With the additional simulation study and reviewer responses, this work was viewed favorably by most reviewers.  We encourage to the authors to incorporate insights from the simulation study into the manuscript.